

# Formation of Highly Oxygenated Low-Volatility Products from Cresol Oxidation

Rebecca H. Schwantes[1], Katherine A. Schilling[2,4], Renee C. McVay[2,5], Hanna Lignell[2,6], Matthew M. Coggon[2,5], Xuan Zhang[1,7], Paul O. Wennberg[1,3], and John H. Seinfeld[2,3]

[1]Division of Geological and Planetary Sciences, California Institute of Technology, Pasadena, California 91125, United States
[2]Division of Chemistry and Chemical Engineering, California Institute of Technology, Pasadena, California 91125, United States
[3]Division of Engineering and Applied Science, California Institute of Technology, Pasadena, California 91125, United States
[4]Current Affiliation: Chemistry and Firearms Branch, US Army Criminal Investigation Laboratory, Forest Park, Georgia, United States
[5]Current Affiliation: Cooperative Institute for Research in Environmental Science and National Oceanic and Atmospheric Administration, Boulder, Colorado, United States.
[6]Current Affiliation: South Coast Air Quality Management District, Diamond Bar, California, United States
[7]Current Affiliation: Aerodyne Research, Billerica, Massachusetts, United States.

*Correspondence to:* Rebecca H. Schwantes (rschwant@caltech.edu)

**Abstract.** Hydroxyl radical (OH) oxidation of toluene produces the ring-retaining products cresol and benzaldehyde, and the ring-opening products bicyclic intermediate compounds and epoxides. Here, first- and later-generation OH oxidation products from cresol and benzaldehyde are identified in laboratory chamber experiments. For benzaldehyde, first-generation ring-retaining products are identified, but later-generation products are not detected. For cresol, low-volatility (saturation mass concentration, C* $\sim$3.5 x $10^4$ - 7.7 x $10^{-3}$ $\mu$g m$^{-3}$) first- and later-generation ring-retaining products are identified. Subsequent OH addition to the aromatic ring of $o$-cresol leads to compounds such as hydroxy, dihydroxy, and trihydroxy methyl benzoquinones and dihydroxy, trihydroxy, tetrahydroxy, and pentahydroxy toluenes. These products are detected in the gas phase by chemical ionization mass spectrometry (CIMS) and in the particle phase using offline direct analysis in real time mass spectrometry (DART-MS). Our data suggest that the yield of trihydroxy toluene from dihydroxy toluene is substantial. While an exact yield cannot be reported as authentic standards are unavailable, we find that a yield for trihydroxy toluene from dihydroxy toluene of $\sim$0.7 (equal to the yield of dihydroxy toluene from $o$-cresol) is consistent with experimental results for $o$-cresol oxidation under low-NO conditions. These results suggest that even though the cresol pathway accounts for only $\sim$20% of the oxidation products of toluene, it is the source of a significant fraction ($\sim$20-40%) of toluene secondary organic aerosol (SOA) due to the formation of low-volatility products.

## 1 Introduction

Aromatic compounds are emitted from both anthropogenic (e.g., solvent use and motor vehicle exhaust) and natural (e.g., wildfires) processes. Oxidation of aromatic compounds leads to the formation of ozone ($O_3$) and secondary organic aerosol (SOA) (Calvert et al., 2002, and references therein). Despite the number of studies performed, the spectrum of gas-phase



aromatic oxidation products remains incomplete, especially those of later generation and those responsible for producing secondary organic aerosol. Toluene, one of the principal aromatic compounds present in the atmosphere, is emitted from both anthropogenic processes (∼60%) and biofuel/biomass burning (∼40%) (Henze et al., 2008). Chamber studies have measured particularly high SOA mass yields (0.9-1.6 $\mu$g/$\mu$g) from toluene (Zhang et al., 2014) when correcting for vapor wall loss using

the Statistical Oxidation Model. Modeling studies, using SOA yields that do not account for vapor wall loss (e.g., 0.1-0.3 $\mu$g/$\mu$g, (Ng et al., 2007)), estimated that toluene SOA contributes ∼4% of the total SOA produced globally (Henze et al., 2008). Incorporation of the updated SOA yields is expected to increase the calculated significance of toluene to the global SOA budget.

Hydroxyl radical (OH) oxidation of toluene takes place via four pathways, yielding benzaldehyde, cresol, bicyclic interme-

diates, and epoxides (Figure 1). Identification of subsequent gas-phase oxidation products from the benzaldehyde and cresol pathways is the focus of this work. These pathways lead to high yields of ring-retaining products. If sustained during subsequent oxidation, these ring-retaining compounds are likely to lead to SOA. Since OH addition to the aromatic ring of toluene increases the reaction rate constant for subsequent OH addition (Calvert et al., 2002), this chemistry accelerates the path to highly oxidized products.

Benzaldehyde forms as a result of hydrogen abstraction from the methyl group of toluene. Reported benzaldehyde yields from toluene oxidation are relatively consistent in the range of 0.053-0.12 (Calvert et al., 2002, and references therein). MCM v3.3.1 recommends a yield of 0.07, which is in the middle of this range (Jenkin et al., 2003; Bloss et al., 2005).

Cresol is produced from OH addition to the aromatic ring of toluene with subsequent $O_2$ addition and $HO_2$ elimination. Measured yields of cresol from toluene oxidation range from 0.03 to 0.385 (Calvert et al., 2002, and references therein) with

several studies converging to a yield of 0.18 (Klotz et al., 1998; Smith et al., 1998). A recent theoretical study suggests a cresol yield of 0.32 (Wu et al., 2014). Cresol yields from OH oxidation of toluene are difficult to measure quantitatively because cresol is prone to losses (e.g., to sampling tubing) that are dependent on the measurement technique (Klotz et al., 1998). Once formed, cresol ($k_{OH}$ ∼5 x $10^{-11}$ cm$^3$ molec$^{-1}$ s$^{-1}$) reacts much faster with OH than its precursor toluene ($k_{OH}$ = 6 x $10^{-12}$ cm$^3$ molec$^{-1}$ s$^{-1}$) (Calvert et al., 2002). Nakao et al. (2012) detected products in the particle phase indicative of successive

OH addition to the aromatic ring of $o$-cresol (i.e., $C_7H_8O_4$ and $C_7H_8O_5$) and phenol and estimated that the cresol pathway contributes ∼20% of SOA produced from toluene. Most studies (Olariu et al., 2002; Caralp et al., 1999) have focused on monitoring first-generation products from cresol and benzaldehyde in the gas-phase, but not second- and third- generation products. The goal of this work is to identify gas-phase pathways and specific oxidization products important for toluene SOA formation by monitoring later-generation products in the gas phase and linking these products to those detected in the particle

phase.





## 2 Methods

Chamber experiments were performed to study products from toluene OH oxidation under both low- and high-NO condi-tions. In order to explore later-generation chemistry and identify important precursors for SOA, later-generation ring-retaining products were also used as the initial precursor.

### 2.1 Experimental Design

All experiments were performed in the 24 m$^3$ Teflon chambers at the Caltech dual chamber facility. Low- and high-NO experiments were carried out in separate chambers to avoid contamination of NO and related compounds in the low-NO chamber. The chambers were flushed with purified air for 24 h prior to each experiment. Purified air is generated by removing volatile organic carbon, ozone, nitrogen oxides, and water vapor from compressed air. Experiments oxidizing toluene, *o*-cresol, 3-methyl catechol, and benzaldehyde under low- and high-NO conditions were performed (Table 1). For all experiments, hydrogen peroxide ($H_2O_2$) as an OH source was injected first by flowing purified air through a glass bulb heated to 36°C; 2 ppm of $H_2O_2$ was used for all gas-phase and high-NO particle-phase experiments, and 4 ppm $H_2O_2$ was used for low-NO particle-phase experiments.

After addition of the oxidant, the volatile organic compound (VOC) was injected. Toluene (99.8% purity) and benzaldehyde ($\geq$ 99% purity) were injected into a glass bulb using a gas-tight syringe. Purified air was passed into the glass bulb and subsequently injected into the chamber at 5 L min$^{-1}$. A weighed amount of *o*-cresol (99.5% purity) was heated to 49°C, and an excess amount of 3-methyl catechol (98% purity) was heated to 36°C while purified air was passed into a glass bulb. A water bath was used to provide consistent heating.

For high-NO experiments, NO (501 ppm in $N_2$, Scott Specialty Gases) was injected into the chamber using a calibrated mass flow controller at the start of the experiment, and continuously throughout the experiment. The goal of the continuous NO injection was to control the amount of NO present during the experiment, such that the level of $NO_x$ remained as low as possible. A kinetic model is used to verify that these experimental conditions are relevant to the atmosphere (see Section 2.3).

For experiments in which particle-phase sampling was performed, the last step included atomization of 0.06 M ammonium sulfate through a $^{210}$Po neutralizer and into the chamber. Photooxidation ($j_{NO2} = 4.4 \times 10^{-3}$ s$^{-1}$) was initiated at least 1 h after all injections were complete to ensure adequate mixing. Because $NO_3$ forms in the chamber and reacts rapidly with a number of compounds present, lights remained on to ensure photolysis of $NO_3$ until all filters had been collected.

Some studies (Tan et al., 2009; Lim et al., 2010) have implicated glyoxal, an OH oxidation product of toluene, in SOA formation under humid conditions, and one study suggested that glyoxal leads to enhanced SOA growth by increasing OH concentrations rather than directly forming aerosol (Nakao et al., 2012). In the present study, all experiments were carried out under dry conditions (RH < 10%) to simplify gas-phase measurements and to focus on the later-generation low-volatility products that form in the gas phase and partition to the particle phase.

In experiment 9, all procedures were the same as described in the proceeding paragraphs, but after 1.5 h of photooxidation, lights were turned off. While lights were off, the decay of 3-methyl catechol oxidation products due to wall deposition was





measured. In experiment 10, all procedures were the same as described above, but lights were turned on for only 3.2 h. Once an adequate level of oxidation products from 3-methyl catechol oxdiation was generated, the chamber experiment was ended and purified air was sampled by the CIMS to monitor the desorption of 3-methyl catechol oxidation products off the CIMS walls.

## 2.2 Chamber Instrumentation

5 Commercial instruments were used to monitor toluene, nitrogen oxides ($NO_x$), ozone ($O_3$), relative humidity (RH), and temperature. Toluene was monitored by a gas chromatograph with a flame ionization detector (GC-FID, Agilent 6890N, HP-5 column). $NO_x$ and $O_3$ were monitored by a Teledyne T2OO $NO_x$ monitor and Horiba APOA-360 $O_3$ monitor, respectively. A Vaisala HMM211 probe was used to monitor temperature and RH. Gas-phase oxidized compounds were detected via a $CF_3O^-$ Chemical Ionization Mass Spectrometer (CIMS) (Section 2.2.1). Particle-phase compounds were monitored using 10 high-resolution direct analysis in real time mass spectrometry (DART-MS) from filters collected at the end of each experiment (Section 2.2.2).

### 2.2.1 CIMS Description and Calibration

A chemical ionization mass spectrometer (CIMS) was used to monitor oxidized organic compounds in the gas-phase. The CIMS uses a custom-modified triple quadrupole mass analyzer (Varian 1200) (St. Clair et al., 2010). The instrument was 15 operated in both negative and positive mode using $CF_3O^-$ and $H_3O \cdot (H_2O)_n^+$, respectively, as the reagent ions. A compound (A) with an affinity for fluorine interacts with $CF_3O^-$ either to form a complex (R1) or a $F^-$ transfer reaction (typically acidic compounds, R2). A compound is detected at its molecular weight + 85 for the complex and + 19 for the $F^-$ transfer. In positive mode, $H_3O^+$ typically interacts with a compound along with $0 - n$ water molecules to form a complex at the molecular weight $+(18n + 1)$ (R3). Other ions (e.g., $NO^+$) also cluster in positive mode complicating interpretation of signals. The reactions 20 are:

$$A + CF_3O^- \rightarrow CF_3O^- \cdot A \tag{R1}$$

$$A + CF_3O^- \rightarrow CF_2O + A_{(-H)}^- \cdot HF \tag{R2}$$

$$25 \quad A + H^+ \cdot nH_2O \rightarrow A \cdot H^+(H_2O)_n \quad n = 0, 1, 2, ... \tag{R3}$$

More detail about the ion chemistry of the CIMS is provided in St. Clair et al. (2010), Crounse et al. (2006), and Paulot et al. (2009). Positive mode was used to monitor the decay of benzaldehyde, which is not detected in negative mode. Negative mode was normalized by the total number of reagent ions. Signals were not normalized for positive mode, because the total reagent ions ($H_3O^+$ and its water clusters) cannot be monitored.



MS/MS mode was used to confirm the identity of certain products and to separate isobaric compounds. In MS/MS mode, only complex interactions will produce a $CF_3O^-$ daughter (m/z = (-)85). Transfer reactions produce an $A^-_{(-H)}$ daughter (m/z = molecular weight of the analyte - 1). Detection of the $A^-_{(-H)}$ daughter and not the $CF_3O^-$ daughter confirms the ion is correctly assigned as a $F^-$ transfer and the analyte is acidic. The structural information provided by MS/MS mode helps correctly

identify compounds.

The CIMS was calibrated using *o*-cresol. An excess amount of *o*-cresol was heated at 46°C in a glass bulb. $N_2$ was blown into this glass bulb and then directed into a Teflon pillow bag to produce a concentrated mixture containing ∼80 ppm of *o*-cresol. A 500 mL glass bulb was filled from this concentrated bag and Fourier transform infrared absorption (FT-IR) spectroscopy (pathlength 19 cm) was used to determine the concentration. After confirmation with FT-IR each time, the remaining *o*-cresol

contained in the glass bulb was used to create a dilute pillow bag (∼200 ppb). The dilute pillow bag was filled with either dry $N_2$ or the same purified air used to fill the large 24 m$^3$ Teflon chambers. This dilute pillow bag was then sampled by the CIMS.

The *o*-cresol integrated cross section for region 3145-2824 cm$^{-1}$ measured by Etzkorn et al. (1999) was used for quantification. To our knowledge there are no other reported FT-IR quantifications of *o*-cresol. *m*-Cresol has been quantified at Pacific Northwest National Laboratory (PNNL) using FT-IR (Sharpe et al., 2004). As verification of the Etzkorn et al. (1999)

calibration, the integrated cross section for the region 3178-2706 cm$^{-1}$ of *m*-cresol was used to evaluate the PNNL spectrum. The PNNL calibration (1 ppm) is 28% lower than the Etzkorn et al. (1999) calibration (1.39 ppm). The absorption spectra for *o*-cresol and *m*-cresol in this region only partially align, but the integrated cross sections measured by Etzkorn et al. (1999) are similar (12.7 x 10$^{-18}$ and 12.6 x 10$^{-18}$ cm molec$^{-1}$, respectively).

Sequential FT-IR runs confirmed loss of *o*-cresol to the glass cell (∼8% in the first 10 min and ∼24% after ∼1 h). Within

10 min of the FT-IR sample collection, the glass bulb was flushed into the dilute pillow bag. If wall deposition of *o*-cresol is reversible, the *o*-cresol that deposited on the wall would be flushed into the pillow bag. Because the extent of reversibility of *o*-cresol wall loss is unknown, a correction for wall loss was not applied, but instead added as uncertainty (8%). Thus, the uncertainty for the *o*-cresol sensitivity is estimated as 36%, a combination of the uncertainty in the FT-IR quantification and loss of *o*-cresol during the calibration.

Traditionally, an analyte (A) is detected either at the $F^-$ transfer reaction (A+19) or complex formation (A+85). However, fragmentation products have also been detected (Praske et al., 2015). *o*-cresol predominantly forms a complex with $CF_3O^-$. The proportion of *o*-cresol that undergoes a transfer reaction versus fragmenting is dependent on the water mixing ratio. Fragmentation is higher in the purified air versus the drier nitrogen (Table 2). Likely the presence of water destabilizes the molecular ion formed from $CF_3O^-$ ionization leading to more fragmentation.

Many of the fragmentation products are small and not uniquely formed from one m/z, and so cannot be used to determine the concentration of an individual compound. Instead all possible unique fragments were considered in determining the concentration of a compound. This includes reactions R1, R2 and the following:

$$A + CF_3O^- + M \rightarrow CF_2O \cdot A^-_{-H} + HF + M \tag{R4}$$





$$A + CF_3O^- + M \rightarrow A^-_{-H} + CF_2O + HF + M \tag{R5}$$

$$A + CF_3O^- + M \rightarrow HF{\cdot}A^-_{-CO2} + CF_2O + CO_2 + M \tag{R6}$$

Because the small fragment signals cannot be uniquely assigned to a specific compound, the fraction of these signals to the total needs to be known for all water levels used in the experiments. The influence of water on the fraction of the $o$-cresol

signal produced from unique signals was determined by sampling a sustained amount of $o$-cresol and sequentially adding more water to the CIMS sampling inlet. For $o$-cresol, the sum of all signals (unique and small fragmentation products) is relatively consistent for the relative humidities used in these experiments (Table 1). The CIMS sensitivity determined from the FT-IR dry $N_2$ calibration was corrected for the influence of water in the purified air. The FT-IR purified air calibration was within 10% of this approach. The water correction for $o$-cresol is minor. The CIMS sensitivity (including only unique signals) decreases by <

1% due to the slight RH change over the course of the experiments.

3-methyl catechol calibration was attempted using the same FT-IR method as $o$-cresol. However, because the vapor pressure of 3-methyl catechol ($6.8 \times 10^{-6}$ atm) is much lower than that of $o$-cresol ($3.9 \times 10^{-4}$ atm) (Table 3), preparation of a sufficiently concentrated pillow bag for FT-IR quantification was not possible. Instead, the sensitivities of $o$-cresol and 3-methyl catechol were assumed to be the same in dry $N_2$ when including the sum of all detected signals (i.e., transfer, complex, and

fragments) with a correction for the difference in the ion-molecule collision rate for the compounds. The ion-molecule collision rate (dependent on the molecular weight, dipole moment, and polarizability of two colliding molecules) was estimated using the technique explained in Su and Chesnavich (1982) (see Section S1 and Tables S1 and S2 of the supplemental information for more details).

The additional OH group on 3-methyl catechol increases the acidity such that it dominantly undergoes a $F^-$ transfer reaction

(Table 2). Unlike complex interactions, $F^-$ transfer reactions are increasingly likely to decompose into smaller fragments as the mixing ratio of water increases (Table 2). The influence of water vapor on the sensitivity of 3-methyl catechol was measured in the same manner as that of $o$-cresol. Unlike $o$-cresol, the sum of all unique signals and small fragmentation products for 3-methyl catechol is not consistent for the relative humidities used in these experiments. Likely at higher water concentrations, more fragmentation products with an m/z <50 (the lower limit of the CIMS scanning range) are produced. The sensitivity

(including only unique signals) decreased over the course of the experiments more for 3-methyl catechol (9-15%) than for $o$-cresol (<1%).

Because the $CF_3O^-$ chemical ionization process for 3-methyl catechol exhibits more fragmentation and dependence on water than $o$-cresol, extrapolating the sensitivities to other more oxidized compounds (e.g., trihydroxy toluene) has a high degree of uncertainty. The fragmentation and water dependence could exceed that for 3-methyl catechol. No authentic standards for

trihydroxy toluene are currently available. However, two isomers (5-methyl-benzene-1,2,3-triol and 2,4,6-trihydroxytoluene) of trihydroxy toluene from Sigma's "collection of rare and unique chemicals" are available. Because Sigma does not validate





the identity and purity of these compounds, these compounds were used only to examine the ion chemistry on the CIMS. Purified air was flowed through a heated ($\sim$60-150°C) glass bulb containing each compound into a Teflon pillow bag. Due to the low volatility (saturation mass concentration, C* $\sim$ 340 $\mu$g m$^{-3}$) of these compounds, introducing detectable amounts into the gas phase without decomposition was extremely difficult.

2,4,6-trihydroxy toluene seemed to be more stable and a higher signal was achieved compared to 5-methyl-benzene-1,2,3-triol. Only major signals for 5-methyl-benzene-1,2,3-triol were above the noise and reported. For 2,4,6-trihydroxy toluene in MS/MS mode, m/z (-)159 produced the m/z (-)139 daughter but not the m/z (-)115 daughter, and m/z (-)225 produced the m/z (-)205 daughter and only minor amounts of the m/z (-)85 daughter. In MS mode, 2,4,6-trihydroxy toluene produced the following signals m/z (-)225 $\sim$ 205 > 159 > 139 as well as many signals attributed to decomposition products or impurities

(e.g., acetic acid). Although the signal intensity for 5-methyl-benzene-1,2,3-triol was low in MS/MS mode, which is less sensitive than MS mode, the signal intensity was sufficient to verify that m/z (-)159 produces the m/z (-)139 daughter. In MS mode, 5-methyl-benzene-1,2,3-triol produced the following signals m/z (-)205 $\sim$ 159 > 225 > 139 and also produced a number of decomposition products or impurities (e.g., formic acid).

   There was a large array of additional signals measured by the CIMS from these standards. These signals are caused by impu-

rities in the standards, decomposition outside of the CIMS due to heating, and fragmentation inside the CIMS during chemical ionization. When the standards were introduced into the pillow bag at different temperatures, the ratio of these compounds to the m/z (-)159 (trihydroxy toluene) signal was not consistent, suggesting these signals are largely due to impurities or decomposition outside of the CIMS. Fragmentation inside the CIMS during ionization would produce relatively consistent product fractions. Further understanding of the fragmentation occurring inside the instrument for trihydroxy toluene was unattainable

owing to the high signals of impurities and decomposition products.

   The sensitivity (all unique signals) determined for $o$-cresol was assumed to extend directly to the following compounds: methyl hydroxy benzoquinone, methyl nitrophenol, benzoic acid, peroxy benzoic acid, phenyl hydroperoxide, nitrophenol, and dinitrophenol with a correction for the ion-molecule collision rate (Table S1). Similarly, the sensitivity (all unique signals) determined for 3-methyl catechol was assumed to extend directly to trihydroxy toluene, tetrahydroxy toluene, dihydroxy methyl

benzoquinone, and dihydroxy nitrotoluene with a correction for the ion-molecule collision rate (Table S1). To the extent possible, all signals (transfer, complex, and potential unique fragmentation products (R1, R2, R4, R5, and R6)) for these compounds were used to determine their mixing ratio.

   During toluene oxidation, $m$-cresol and $p$-cresol also form. $o$-,$m$-, and $p$-cresol all produce similar amounts of non-unique fragmentation products in purified air (89-91%). Therefore, the slight difference in the ion-molecule collision rate (Table S1)

and the isomer distribution produced during toluene oxidation (Klotz et al., 1998) was used to calculate a general cresol sensitivity.

### 2.2.2 DART-MS Description

SOA was collected during the final 4 h of experiments at 24 L min$^{-1}$ on a Teflon membrane filter (47 mm, 1.0 $\mu$m pore size, Pall Life Sciences). The filters were analyzed by high-resolution direct analysis in real time mass spectrometry (DART-MS,



JEOL, Inc.). A DART source is a low-temperature He plasma that generates primarily $[A+H]^+$ ions through proton transfer reactions between the analyte, A, and ionized ambient water vapor ($H_3O^+$) (Cody et al., 2005; Cody, 2009). Samples are introduced directly into the DART stream, between the end of the DART source and the mass spectrometer inlet. A portion of the filter membrane was cut free from the support ring using a stainless steel scalpel and wrapped in a spiral around the barrel

of a glass Pasteur pipet. The pipet was rotated slowly in the DART stream to warm the glass and desorb organic material gently from the Teflon filter. Each sample was cut and analyzed in triplicate. The final data are an average of these three replicates. Additional analysis details and interpreted mass spectral data corrected to remove background ions are provided in Section S3 of the Supplemental Information.

With such a broad spectrum of compounds and the absence of synthetic standards, only ions with signals well above the

background were selected for analysis. Ions with signals > 10% of the maximum ion signal (experiments 14 and 15) or second maximum ion signal (experiment 13) were selected. In experiment 13, the second maximum signal was used for peak selection instead of the first because the first maximum ion signal dominated the mass spectrum (i.e., >6 times any other ion signal). The accurate m/z of each selected ion was assigned a chemical formula using ChemCalc (Patiny and Borel, 2013). This chemical formula was adjusted to its neutral form, and given a proposed structure based on the Master Chemical Mechanism (MCM)

v3.3.1 (Jenkin et al., 2003; Bloss et al., 2005) toluene photooxidation mechanism, previously reported components of toluene SOA (Calvert et al., 2002; Olariu et al., 2002; Sato et al., 2007; Jang and Kamens, 2001; Nakao et al., 2011), and gas-phase photooxidation products detected here by the CIMS.

DART-generated signal intensity for a given compound is proportional to the product of its vapor pressure, proton affinity, and concentration (Nilles et al., 2009; Schilling Fahnestock et al., 2015; Chan et al., 2013). Because the ion intensity is

proportional to the vapor pressure, the vapor pressure of each compound needs to be known or estimated. Estimates of vapor pressures for low-volatility compounds have higher uncertainty due to lower availability and accuracy of experimental data (Barley and McFiggans, 2010; O'Meara et al., 2014; Kurten et al., 2016). Thus, the results presented for the DART-MS analysis should be interpreted only qualitatively.

Two vapor pressure estimation methods are used here: 1) the Estimation of Vapor Pressure of Organics, Accounting for

Temperature, Intramolecular, and Non-additivity Effects (EVAPORATION) method (Compernolle et al., 2011) and; 2) the method of Nannoolal et al. (2004, 2008). Both methods have online tools available for estimating the vapor pressure at http://tropo.aeronomie.be/models/evaporation_run.htm and http://www.aim.env.uea.ac.uk/aim/ddbst/pcalc_main.php, respectively. The EVAPORATION and Nannoolal methods are compatible with molecules containing oxygen-based functional groups and nitrates. Unlike the Nannoolal method, the EVAPORATION method has not been optimized for aromatic compounds, while

the Nannoolal method cannot be used for diketones. Thus, EVAPORATION is used for all non-aromatic compounds and Nannoolal is used for all aromatics.

## 2.3 Kinetic Model

The chamber experiments were simulated with a kinetic model containing all reactions related to toluene from MCM v3.3.1 (Jenkin et al., 2003; Bloss et al., 2005), via http://mcm.leeds.ac.uk/MCM. Version 1 of the kinetic model includes all MCM





v3.3.1 reactions relevant to toluene oxidation and inorganic chemistry, as well as experimentally measured wall deposition rates of *o*-cresol and dihydroxy toluene. Version 2 includes all reactions in Version 1 as well as photolysis of hydroxy nitrotoluene and dihydroxy nitrotoluene. Version 3 includes all reactions in Version 2 as well as additional oxidation reactions for dihydroxy toluene and benzaldehyde. Additional discussion of the kinetic model, including a list of reactions, is provided in Section S2

of the Supplemental information.

The kinetic model was used to evaluate the extent to which chamber conditions are representative of those in the atmosphere. The two main concerns in chamber studies performed under high-NO conditions are high $NO_2$ and $NO_3$ levels. Upon reaction with OH, a VOC forms an OH-VOC adduct, that will react with either $NO_2$ or $O_2$. Under atmospherically relevant conditions, the OH-VOC adduct reacts predominantly with $O_2$. The $NO_2$ reading on the $NO_x$ monitor used in this study includes all $NO_y$

products (e.g., organic nitrates, $HNO_3$, HONO, and $NO_2$). Instead of using the $NO_x$ monitor, the kinetic model was used to predict the maximum $NO_2$ concentration. OH-*o*-cresol and OH-3-methyl catechol adducts are assumed to react at the same rate with $NO_2$ and $O_2$ as OH-*m*-cresol adduct (Koch et al., 2007), and the OH-benzaldehyde adduct was assumed to react with $NO_2$ and $O_2$ at the same rate as the OH-toluene adduct (Koch et al., 2007). The percent of OH-VOC adduct reacting with $NO_2$ versus $O_2$ for each experiment is presented in Table S5. For gas-phase experiments, the percentage of the OH-VOC

adduct reacting with $NO_2$ was <6%. The higher loading necessary for the filter analysis required larger amounts of $NO_x$ for the particle-phase experiments, for which the percentage of OH-VOC reacting with $NO_2$ was <10%.

Both *o*-cresol (1.4 x $10^{-11}$ cm$^3$ molec$^{-1}$ s$^{-1}$, (Atkinson et al., 1992)) and 3-methyl catechol (1.7 x $10^{-10}$ cm$^3$ molec$^{-1}$ s$^{-1}$, (Olariu et al., 2004)) react rapidly with $NO_3$. For the toluene high-NO experiments, a substantial amount of 3-methyl catechol and *o*-cresol is predicted to react with $NO_3$ (e.g., as much as 80% for the particle-phase experiments, Table S5).

Caution is needed when interpreting results for high-NO oxidation conditions as both $NO_3$ and OH oxidation occur. In the present work, when starting with *o*-cresol or 3-methyl catechol, the percentage reacting with $NO_3$ was minor (e.g., < 4% for *o*-cresol for Experiment 5, Table S5). The kinetic model was also used to verify that $RO_2$ + $RO_2$ reactions were minimized for all experiments (Table S5). For example, in the toluene low-NO experiments, $RO_2$ + $RO_2$ reactions for the gas-phase and particle-phase experiments were estimated to be <12% and <18%, respectively, of all $RO_2$ pathways.

## 3   Results

Toluene reacts with OH to form both ring-retaining products (cresol and benzaldehyde) and ring-opening products (bicyclic intermediate compounds and epoxides) (Figure 1). Later generation gas-phase oxidation products from the ring-retaining pathways are identified. These oxidation products have a range of volatilities (C* ~5 x $10^5$ - 7.7 x $10^{-3}$ $\mu$g m$^{-3}$). Compounds with the lower volatilities are detected in the particle phase, implying that the ring-retaining pathways are important for SOA

formation. In order to monitor later generation products and constrain the pathways from which products emerge, oxidation of first-generation products (*o*-cresol and benzaldehyde) and second-generation products (3-methyl catechol) was performed under both high- and low-NO conditions.



### 3.1  *o*-Cresol Oxidation

Previous studies generally recommend a ~0.18 yield of cresol (total of all isomers) from the toluene + OH pathway (Klotz et al., 1998; Smith et al., 1998) (Figure 1). The kinetic model (Version 1) assuming a 0.18 yield predicts cresol levels within the uncertainties of the CIMS measurements under both low- and high-NO conditions (Figure 2). As noted, version 1 of the kinetic

model includes all MCM v3.3.1 reactions related to toluene and wall deposition (see Section 4.2.1) of *o*-cresol and dihydoxy toluene. An approximate cresol yield (~0.2) was calculated using the equation of Olariu et al. (2002) and the decay of toluene, rise in cresol, and accounting for losses of cresol from wall deposition and reaction with OH. The yield was calculated only under low-NO conditions. Under high-NO conditions, the correction for *o*-cresol reaction with $NO_3$ adds more uncertainty (Figure 2). The yield determined here is similar to that of other studies (Klotz et al., 1998; Smith et al., 1998).

The *o*-cresol oxidation mechanism in MCM v3.3.1, based on Olariu et al. (2002), is shown in black in Figure 3 for low-NO conditions and in Figure 4 for high-NO conditions. OH reacts with *o*-cresol via hydrogen abstraction to form a methyl phenoxy radical or addition to form either a bicyclic intermediate product or dihydroxy toluene (the dominant isomer being 3-methyl catechol). In experiment 4, in which 29 ppb of toluene was oxidized, the maximum detected mixing ratio of dihydroxy toluene was only ~0.2 ppb (Figure 2), emphasizing the importance of starting with later-generation products in order to determine

the subsequent chemistry. Photooxidation of *o*-cresol produces dihydroxy toluene (m/z (-) 143) in agreement with Olariu et al. (2002). Under high-NO conditions, the methyl phenoxy radical reacts with $NO_2$ to form hydroxy nitrotoluene (m/z (-)172, $F^-$ transfer and m/z (-)152, fragment).

3-Methyl catechol oxidation leads to the following products (Figure 3): trihydroxy toluene, hydroxy methyl benzoquinone, and various decomposition products presumably from the bicyclic intermediate pathway. These products likely result from

OH addition to the ring of 3-methyl catechol. This pathway is not included in MCM v3.3.1, which assumes that hydrogen abstraction is the sole OH oxidation pathway for 3-methyl catechol. Dihydroxy nitrotoluene (m/z (-)188, $F^-$ transfer and m/z (-)168, fragment), the expected product of hydrogen abstraction of 3-methyl catechol, is detected minimally (< 0.5 ppb) by the CIMS. This suggests that hydrogen abstraction is not the dominant pathway for OH oxidation.

Without authentic standards for trihydroxy toluene and hydroxy methyl benzoquinone, quantification cannot be achieved

(Section 2.2.1). Exact yields are not reported, but the experimental results are compared to the kinetic model (Section 4). Trihydroxy toluene is detected at several different signals in MS mode on the $CF_3O^-$ CIMS: m/z (-)159 ($F^-$ transfer), m/z (-)225 (complex), m/z (-)115 (fragment, possibly loss of $CO_2$), m/z (-) 205 (loss of HF from complex), and m/z (-) 139 (loss of HF from transfer). Three daughters are detected in MS/MS mode from m/z (-)159: m/z (-)139 (loss of HF), m/z (-)115 (possibly loss of $CO_2$), and m/z (-)85 ($CF_3O^-$). The presence of the m/z (-)85 daughter implies two compounds are detected at m/z (-

)159: trihydroxy toluene and another compound that forms a $CF_3O^-$ complex (e.g., hydroxyacetone). Here MS/MS mode is used to separate the trihydroxy toluene signal from the interfering compound. Hydroxy methyl benzoquinone is detected at m/z (-)223 (complex), m/z (-)157 ($F^-$ transfer), and m/z (-)137 (fragment).

Several products from photooxidation of trihydroxy toluene are also detected by the CIMS in the 3-methyl catechol oxidation experiments, including tetrahydroxy toluene, dihydroxy methyl benzoquinone, and various decomposition products from the



bicylic intermediate pathway (Figure 3). Tetrahydroxy toluene, like trihydroxy toluene, is detected at m/z (-)175 ($F^-$ transfer), m/z (-)241 (complex), and m/z (-)131 (fragment, possibly loss of $CO_2$). Dihydroxy methyl benzoquinone is detected at m/z (-)239 (complex), m/z (-)173 ($F^-$ transfer), and m/z (-)153 (fragment). Trihydroxy methyl benzoquinone (m/z (-)189) and pentahydroxy toluene (m/z (-)191), likely oxidation products from tetrahydroxy toluene, are also detected by the CIMS, but

the signals are close to background. As shown in Figure 3, OH oxidation of methyl benzoquinone possibly also forms products detected at the same mass as pentahydroxy toluene and dihydroxy methyl benzoquinone. However, these products are detected from 3-methyl catechol oxidation consistent with the proposed mechanism (Figure 3).

An array of decomposition products presumably from the bicyclic intermediate oxidation pathway of *o*-cresol, 3-methyl catechol, trihydroxy toluene, and tetrahydroxy toluene is detected (Figures 5 and 6). These decomposition products vary greatly

in volatility (C* $\sim$ 2.2 x $10^6$ to 14 $\mu$g m$^{-3}$). The highly oxygenated products such as $C_4H_4O_5$ and $C_5H_6O_5$ (C* $\sim$ 14 to 41 $\mu$g m$^{-3}$) are likely to result only from trihydroxy toluene and tetrahydroxy toluene oxidation, and are sufficiently low in volatility to partition in some degree to the particle phase. Because theoretical (PengZhen et al., 2012; Xu and Wang, 2013; Jorgensen, 2012) and experimental (Olariu et al., 2002) studies of OH addition to phenol and *o*-cresol all suggest *ortho*-addition is dominant, OH is presumed to also add to the *ortho*-position in Figures 5 and 6. OH addition to the other positions of the

ring produces similar products. For *o*-cresol and 3-methyl catechol, all possibilities of OH addition are enumerated and the additional products appear at the bottom of Figure 5.

### 3.2   Benzaldehyde Oxidation

MCM v3.3.1 recommendations for OH oxidation of benzaldehyde are generally in agreement with the products detected by the CIMS (Figure 7). OH oxidation of benzaldehyde occurs via hydrogen abstraction of the formyl group followed by $O_2$

addition to form a peroxy radical. This peroxy radical reacts with $HO_2$ under low-NO conditions to form benzoic acid, peroxybenzoic acid, and phenyl hydroperoxide. Benzoic acid (m/z (-) 141) and peroxybenzoic acid (m/z (-)223) are the dominant first-generation products detected. Phenyl hydroperoxide (m/z (-)129) is minimally detected (<0.2 ppb) either due to a low yield or instability in the CIMS (Hydroperoxides have been known to fragment in the $CF_3O^-$ CIMS (Praske et al., 2015)). As a result of the relatively large $RO_2$ + $RO_2$ rate constant for the peroxy radical of benzaldehyde (1.1 x $10^{-11}$ cm$^3$ molec$^{-1}$

s$^{-1}$), the benzaldehyde low-NO experiment was characterized by the kinetic model as having a higher fraction of $RO_2$ + $RO_2$ reactions than other experiments (Table S5). The proportions of benzoic acid and peroxybenzoic acid measured by the CIMS differ from those predicted by MCM v3.3.1 (see Section S1 and Figure S2 for more details).

Other first-generation products are also detected, including signals at m/z (-)155, (-)175, and (-)179. These minor signals comprise only 6%, 3%, and 5%, respectively, of the signals produced from benzoic and peroxybenzoic acids. Phenol is likely

at m/z (-)179. Compounds forming signals at m/z (-)155 and (-)175 rise with the other first-generation products, suggesting they are minor first-generation products from the $RO_2$ + $RO_2$ or $RO_2$ + $HO_2$ pathways.

The dominant first-generation product detected from benzaldehyde oxidation under high-NO conditions is nitrophenol (m/z (-)158, $F^-$ transfer and m/z (-)138, fragment) (Figure 7). Dinitrophenol (m/z (-)203, $F^-$ transfer and m/z (-)183, fragment), an



OH oxidation product of nitrophenol, was also detected. Both products are over-predicted by MCM v3.3.1 using the kinetic model (Section 4.2.2 and Figure S3) compared to the CIMS measurements.

OH addition to the aromatic ring of benzaldehyde or benzoic acid is expected to be only a minor pathway. The rate of OH addition to an aromatic ring is proportional to the electrophilic nature of the substituents around the ring; unlike methyl and hydroxy groups, carboxy and formyl groups are not electrophilic (Calvert et al., 2002). OH addition to the ring of benzaldehyde would form hydroxy benzaldehyde, which is isobaric to benzoic acid. Only the transfer signal (m/z (-)141), not the complex (m/z (-)207), is detected, indicative that the product is highly acidic, like a carboxylic acid. That hydroxy benzoic acid does not form cannot be explicitly confirmed because this compound is isobaric to peroxybenzoic acid. However, dihydroxy benzoic acid (m/z (-)173) as expected does not form.

Oxidation of benzaldehyde under high- and low-NO conditions does not yield many later generation products detectable by the $CF_3O^-$ CIMS. The CIMS is expected to be sensitive to later generation products from the benzaldehyde pathway that retain the aromatic ring. Likely the main later generation products from benzaldehyde are ring-opening decomposition products, to which the CIMS is not sensitive (e.g., unfunctionalized ketones and aldehydes). Thus, we conclude that the cresol pathway is more important for SOA formation compared to the benzaldehyde pathway, based on the detectable products of the $CF_3O^-$ CIMS and their expected volatilities (Table 3).

### 3.3 Products Detected in the Particle Phase by DART-MS

Products detected in the gas phase are compared to those detected in the particle phase to further understand the mechanism for toluene SOA formation. Filters, collected at the end of each experiment, were analyzed using high-resolution direct analysis in real time mass spectrometry (DART-MS). As expected, a number of compounds (e.g., trihydroxy toluene, tetrahydroxy toluene, and pentahydroxy toluene) measured in the gas-phase were also detected in the particle-phase by the DART-MS.

The intensity of the DART signal for a given compound is proportional to the product of the proton affinity, vapor pressure, and concentration of the compound. The proton affinity of each compound is assumed to be similar, due to shared ionizable functional groups. To compare the relative amounts of each product detected, the measured intensity is normalized by the compound's estimated vapor pressure to produce a normalized intensity ($I_n$). The relative fraction ($R_f$) of each compound is then calculated by dividing each compound's normalized intensity by the sum of the normalized intensities of all compounds in a given experiment. $R_f$ values for each compound detected are reported in Tables S6-S8 in the Supplemental Information.

Vapor pressures for the compounds detected in this study have been estimated using both EVAPORATION and Nannoolal methods (Compernolle et al., 2011; Nannoolal et al., 2004, 2008). As noted earlier (Section 2.2.2), due to limitations in the methods, Nannoolal is used for all aromatics and EVAPORATION is used for all non-aromatic compounds. As demonstrated in Table 3, as the volatility of an aromatic compound decreases, the EVAPORATION method increasingly underestimates vapor pressures as compared to Nannoolal. Owing to the uncertainty in these vapor pressure estimates, the reported $R_f$ values should be considered only qualitatively.

The same *o*-cresol oxidation products, detected by the CIMS in the gas phase and expected to be low in volatility, are detected in the particle phase by the DART-MS. The corroborative analyses by CIMS and DART-MS support the proposed mechanism





that OH subsequently adds to the ring of *o*-cresol forming low-volatility products. For example, the following are dominant products detected in the particle phase under low-NO oxidation of toluene (Figure 8a): polyols including trihydroxy toluene ($C_7H_8O_3$), tetrahydroxy toluene ($C_7H_8O_4$), and pentahydroxytoluene ($C_7H_8O_5$); benzoquinones including hydroxy methyl benzoquinone ($C_7H_6O_3$) and dihydroxy methyl benzoquinone ($C_7H_6O_4$); and various products from the bicyclic intermediate

pathway including $C_4H_4O_2$, $C_5H_6O_2$, and $C_5H_6O_3$. Nakao et al. (2012) also detected $C_7H_8O_4$ and $C_7H_8O_5$ in the particle phase from *o*-cresol oxidation under low-NO conditions using a Particle-into-Liquid Sampler coupled with a Time-of-Flight Mass Spectrometer. Nakao et al. (2012) suspected these signals were due to successive OH addition to the aromatic ring; the combined CIMS and DART-MS analysis corroborates their conjecture.

As shown in Figure 8b, similar products were detected under toluene high-NO oxidation. Under high-NO conditions, methyl

nitrophenol and dihydroxy nitrotoluene were also detected in the particle phase, albeit at lower relative fractions than many of the polyols and benzoquinone compounds (Figure 8b). At the end of the toluene high-NO experiment, estimated $NO_3$ oxidation of *o*-cresol (80%) and 3-methyl catechol (66%) was quite high compared to OH oxidation (Table S5). Part of the methyl nitrophenol signal could be influenced by $NO_3$ oxidation.

Some of the compounds (e.g., tetrahydroxy toluene and pentahydroxy toluene) are structural isomers of those produced from

the epoxide pathway of toluene oxidation under low-NO conditions; under high-NO conditions, the products from the epoxide channel largely decompose (Figure S6). Signals assigned to tetrahydroxy toluene and pentahydroxy toluene are dominant in the particle phase from toluene oxidation under both high- and low-NO conditions (Figure 8). This is consistent with the products forming from the *o*-cresol oxidation mechanism proposed in Figure 3 rather than the epoxide mechanism. Hydroxy methyl hydroperoxy benzoquinone, a product of OH oxidation of benzoquinone under low-NO conditions, and pentahydroxy

toluene are structural isomers (Figure 3). Because the signal assigned to pentahydroxy toluene is detected under both low- and high-NO conditions, the product is more likely to be generated from tetrahydroxy toluene than benzoquinone.

Nitrophenol from benzaldehyde oxidation is also detected in the particle phase. Part of the signal for $C_7H_6O_3$ may represent peroxybenzoic acid, as this compound is isobaric to hydroxy methyl benzoquinone. In general, products from cresol oxidation dominate over products from benzaldehyde oxidation in the particle phase. This is consistent with the gas-phase chemistry

discussed in Section 3.2, in which few ring-retaining later generation products are detected from benzaldehyde oxidation.

### 3.4 Estimating the Contribution of Cresol to Toluene SOA

Considering that many products produced by the cresol pathway in the gas phase are also detected in the particle phase, the contribution of these products to toluene SOA is estimated. Chamber studies have recently reported toluene SOA mass yields to be between 0.9-1.6 $\mu g/\mu g$ when using the Statistical Oxidation Model to correct for vapor wall loss (Zhang et al., 2014).

Without the model corrections for vapor wall loss, Zhang et al. (2014) measured the toluene SOA mass yields to be 0.5-0.65 $\mu g/\mu g$ for the experiments at maximum seed aerosol surface area (i.e., the experiments that minimized vapor wall loss as much as possible).

Under low-NO conditions the toluene SOA yield with the model corrections for vapor wall loss (1.6 $\mu g/\mu g$) implies a near unity carbon yield (Zhang et al., 2014), so at the lower bound the cresol pathway contributes to ∼20% of toluene SOA. For





an upper bound estimate, we assume that trihydroxy toluene and hydroxy methyl benzoquinone and all oxidation products therefrom partition to the particle phase and that the average molecular weight of all compounds in the aerosol is equal to that of pentahydroxy toluene. With these assumptions and using the toluene SOA mass yield of 0.5 (the lowest yield explained above), the cresol pathway is estimated to contribute ~40% of toluene SOA. Based on this, the contribution of the cresol

pathway to toluene SOA is estimated as ~20-40%.

## 4   Discussion

Gas- and particle-phase measurements by the CIMS and DART-MS confirm that OH oxidation of dihydroxy toluene leads to low-volatility products that partition to the particle phase. For example, the following three products, which form from subsequently adding OH to the aromatic ring, are detected in the gas and particle phases: trihydroxy toluene (C* ~ 340 $\mu$g

m$^{-3}$), tetrahydroxy toluene (C* ~ 2.1 $\mu$g m$^{-3}$) and pentahydroxy toluene (C* ~ 7.7 x 10$^{-3}$ $\mu$g m$^{-3}$). Here, we discuss other theory and experimental work pertaining to OH addition to the aromatic ring and use the kinetic model to further interpret the CIMS results.

### 4.1   OH Addition to an Aromatic Ring

The chemical mechanism proposed in Figure 3 is consistent with previous observations of aromatic chemistry. As OH groups

add to an aromatic ring, the aromatic ring becomes more activated, and the OH addition rate constant increases (Calvert et al., 2002). For example, the OH reaction rate constants for toluene, $o$-cresol, and 3-methyl catechol are 6 x 10$^{-12}$, 4 x 10$^{-11}$, and 2 x 10$^{-10}$ cm$^3$ molec$^{-1}$ s$^{-1}$, respectively (Calvert et al., 2002; Olariu et al., 2000).

In the atmosphere, once OH adds to the aromatic ring, O$_2$ also adds, and one of the following four processes occurs: O$_2$ elimination, HO$_2$ elimination, bicyclic intermediate formation, or reaction with NO, HO$_2$ or RO$_2$. For the toluene system,

HO$_2$ elimination to re-form the aromatic ring or cyclization to form a bicyclic intermediate are the most commonly expected processes. Experiment and theory both suggest that HO$_2$ elimination occurs more rapidly than bicyclic intermediate formation for products with more hydroxy functional groups around the aromatic ring. Experimental studies report the cresol yield from toluene to be ~0.2 (Klotz et al., 1998; Smith et al., 1998), while the 3-methyl catechol yield from $o$-cresol is ~0.7 (Olariu et al., 2002). By analogy, the yield of trihydroxy toluene, the OH addition product from 3-methyl catechol, is also expected to

be substantial. Theoretical calculations for phenol suggest that the elimination of HO$_2$ after O$_2$ addition occurs faster than the formation of a bicyclic intermediate (Xu and Wang, 2013). The O$_2$ addition reaction rate constant for the phenol-OH adduct at 323 K (300 x 10$^{-16}$ cm$^3$ molec$^{-1}$ s$^{-1}$ (Koch et al., 2007)) is suggested by Xu and Wang (2013) to be large due to the rapid elimination of HO$_2$. In contrast, the O$_2$ addition rate constant for the toluene-OH adduct at 321K (5.6 x 10$^{-16}$ cm$^3$ molec$^{-1}$ s$^{-1}$ (Koch et al., 2007)) is much lower. Similarly, HO$_2$ elimination may pull the O$_2$ reaction channel forward in the cresol

system as the O$_2$ addition rate constant for $m$-cresol-OH adduct (800 x 10$^{-16}$ cm$^3$ molec$^{-1}$ s$^{-1}$ (Zetzsch et al., 1997)) is even larger than that of phenol.





## 4.2 Comparing CIMS Measurements with Kinetic Model Predictions

In order to evaluate the products detected by the CIMS, the mechanism outlined in Figures 3-7 is incorporated into the kinetic model. Version 2 of the kinetic model includes photolysis of hydroxy nitrotoluene and dihydroxy nitrotoluene. Version 3 of the kinetic model includes additional products for 3-methyl catechol and benzaldehyde oxidation (see Section S2 for more details).

### 5    4.2.1    Vapor Wall Deposition

In order to compare the CIMS results to the kinetic model predictions, all loss processes need to be constrained for the compounds of interest. This includes reaction with OH/NO$_3$ and vapor wall deposition. Considering the reaction rate constant for 3-methyl catechol + OH is already fast (2.0 x 10$^{-10}$ cm$^3$ molec$^{-1}$ s$^{-1}$) (Olariu et al., 2000), the hard sphere collision rate limit (2.5 x 10$^{-10}$ cm$^3$ molec$^{-1}$ s$^{-1}$) is assumed for OH reaction with trihydroxy and tetrahydroxy toluene. The approximations for NO$_3$ oxidation are more speculative than those for OH oxidation, so more focus is given to comparing the CIMS measurements and the kinetic model results under low-NO conditions.

In experiment 9 (Table 1), lights were turned on to generate oxidation products of 3-methyl catechol. After 1.5 h, lights were turned off, and wall deposition rates of the following five compounds were measured: 3-methyl catechol, trihydroxy toluene, tetrahydroxy toluene, hydroxy methyl benzoquinone, and dihydroxy methyl benzoquinone. After 9 h of lights off at 28°C, the chamber was heated to 42°C for 4 h, then cooled to 16°C for 3 h, and finally heated back to 28°C for 3 h (Figure 9). An equilibrium is established for each compound between the gas phase and chamber wall. Heating and cooling the chamber disrupts this equilibrium. Most of the compounds did not significantly re-partition back into the gas phase when heating the chamber from 28°C to 42°C. This implies the absence of a large reversible reservoir on the chamber walls, so measuring the decay after 1.5 h of photooxidation is reasonable for these compounds. All of the compounds were lost to the walls faster at 16°C than at 28°C. As the chamber was heated from 16 to 28°C, some of the products slightly re-partitioned back to the gas phase, but not to the level expected if the chamber had not been cooled, suggesting that some loss was irreversible.

In experiment 10 (Table 1), photooxidation products from 3-methyl catechol were generated in the chamber. After 3.2 h of oxidation, purified air was sampled on the CIMS to monitor desorption of these oxidation products off the walls of the instrument. Desorption was minimal (i.e., within two scans, the signal was <0.08 of the original signal) for all of the five compounds measured except tetrahydroxy toluene (Figure S1). The signals for tetrahydroxy toluene dropped to ∼1/2 their original value after sampling purified air, and slowly decayed from this point onward. Therefore, this compound is lost reversibly to the walls of the CIMS such that accurate quantification and wall loss determination is not possible. This was the only compound that rose nearly to its original signal after heating the chamber to 42°C (Figure 9b, cyan), suggesting a possible large reversible reservoir of this compound on the wall as well.

For 3-methyl catechol (m/z (-)143), hydroxy methyl benzoquinone (m/z (-)157), and dihydroxy methyl benzoquinone (m/z 173), MS mode was used to determine the wall deposition rate constants (Figure 9b) (MS/MS mode produced similar results). The MS and MS/MS results for trihydroxy toluene (Figure 9a) imply that multiple isomers, with different wall loss rates and fragmentation patterns on the CIMS, are forming. For example, the wall loss rate determined from the MS signal m/z (-) 159





(2.3 x $10^{-5}$ s$^{-1}$) is different from those determined from the MS/MS daughters of m/z (-)159: 1.8 x $10^{-5}$ s$^{-1}$ (m/z (-) 159), 1.9 x $10^{-5}$ s$^{-1}$ (m/z (-)139), and 3.6 x $10^{-5}$ s$^{-1}$ (m/z (-)115).

The pKa values of compounds similar to trihydroxy toluene demonstrate that aromatic compounds with an OH group *ortho* to another OH group are more acidic than those with an OH group *meta* or *para* to another OH group. For example, 1,2 dihydroxy benzene, 1,3 dihydroxy benzene, and 1,4 dihydroxy benzene have the following pKa values 9.36, 9.44, and 9.91, respectively (Dean, 1992). Likely the arrangement of the OH groups on trihydroxy toluene influences the acidity, which then influences both the wall deposition and the CIMS fragmentation patterns. The two standards of trihydroxy toluene monitored on the CIMS (Section 2.2.1) also demonstrate this effect. The isomer with the hydroxy groups spread out among the ring (2,4,6-trihydroxytoluene) was detected more at the complex than 5-methyl-benzene-1,2,3-triol, implying it is less acidic than 5-methyl-benzene-1,2,3-triol.

For simplicity, the wall deposition rate determined for m/z (-)159 is used in the kinetic model to represent the wall deposition of all trihydroxy toluene isomers, but in order to understand more fully the yield of this product, the isomer distribution and wall deposition for each isomer would need to be known. The water curve correction was applied in calculating the wall deposition rate constants for *o*-cresol, 3-methyl catechol, and trihydroxy toluene, but not for hydroxy methyl benzoquinone or dihydroxy methyl benzoquinone as the influence of water on benzoquinones is unknown. The *o*-cresol and 3-methyl catechol water curve corrections add 2 x $10^{-8}$ and 1.6 x $10^{-6}$ s$^{-1}$ uncertainty, respectively, to the measured wall deposition rate constants. The wall deposition for tetrahydroxy toluene was approximated in the kinetic model since it cannot be measured. A plot of the natural log of C* versus the natural log of the wall deposition rate constant produces a fairly linear fit for the compounds measured in this work (Figure S5). This fit is used to approximate the wall loss of tetrahydroxy toluene, pentahydroxy toluene, and trihydroxy methyl benzoquinone. There is more uncertainty in the wall deposition rates for these compounds due to the necessity of extrapolation.

### 4.2.2 Formation and Loss of Nitro and Nitroso Compounds

MCM v3.3.1 does not include the photolysis of many nitro compounds, even though recent studies have measured fast photolysis rates (Bardini, 2006; Bejan et al., 2007, 2006). For example, hydroxy nitrotoluene has an atmospheric lifetime with respect to photolysis of < 1 h (Bejan et al., 2007). No studies thus far have reported the photolysis rate constants of dihydroxy nitrotoluene. In version 2 of the kinetic model, the photolysis rate constant for 6-methyl-2-nitrophenol (Bejan et al., 2007) was used for hydroxy nitrotoluene and dihydroxy nitrotoluene with a correction for the difference in the NO$_2$ photolysis rate constants between the Caltech chamber and that used by Bejan et al. (2007).

Under high-NO conditions during 3-methyl catechol oxidation, dihydroxy nitrotoluene is detected only minimally (< 0.5 ppb) even though the MCM v3.3.1 chemical mechanism predicts a peak value of ~60 ppb (Figure 3 and Figure 10). Even after accounting for possible photolysis of dihydroxy nitrotoluene, the kinetic model (Version 2) still predicts a peak value of ~45 ppb. Other nitro compounds are detected quite well by the CIMS (e.g., hydroxy nitrotoluene (Section 3.1) and nitrophenol (Section 3.2)), so this is unlikely caused by a sensitivity issue. Dihydroxy nitrotoluene has a lower estimated vapor pressure than the other nitro compounds detected (Table 3), so this compound is possibly lost to chamber walls and Teflon tubing to a


higher degree. Considering the estimated photolyis rate constant ($1.73 \times 10^{-4}$ s$^{-1}$) is nearly an order of magnitude larger than the predicted wall deposition rate constant ($1 \times 10^{-5}$ s$^{-1}$, Figure S5), losses to chamber walls and Teflon tubing are unlikely the explanation for minimal detection of this compound. More likely, dihydroxy nitrotoluene, the expected product of hydrogen abstraction of 3-methyl catechol, is not, in fact, a main product. Contrary to the recommendations of MCM v3.3.1, hydrogen

abstraction contributes only a small degree to OH oxidation of 3-methyl catechol.

MCM v3.3.1 predictions of nitrophenol, a product of benzaldehyde oxidation, exceed the CIMS measurements (Figure S3). An estimated photolysis rate constant was added to the kinetic model based on that for 2-nitrophenol measured by Bardini (2006) and reported by Chen et al. (2011). This photolysis rate constant was corrected for the difference in the photolysis of NO$_2$ between the Caltech chamber and the atmosphere when/where Bardini made the measurement. With this adjustment

(Figure S3), nitrophenol is under-predicted by the kinetic model compared to the CIMS measurements, but within uncertainties in the CIMS sensitivity and the photolysis approximation.

### 4.2.3   *o*-Cresol Oxidation Products

During *o*-cresol oxidation, theory suggests that OH addition occurs dominantly *ipso* (position 2) or *ortho* (position 1 and 3) to the OH substituent due to hydrogen bonding (Jorgensen, 2012). Only OH addition to position 3 will form a dihydroxy

toluene, as positions 1 and 2 do not have an abstractable hydrogen to undergo HO$_2$ elimination. Consistent with these theory calculations, Olariu et al. (2002) detected only 3-methyl catechol from *o*-cresol oxidation. Here, results suggest that 3-methyl catechol is not the only isomer of dihydroxy toluene that forms from *o*-cresol oxidation. 3-methyl catechol is detected mostly at the F$^-$ transfer (97%) and only minimally at the cluster (3%). However, dihydroxy toluene produced from *o*-cresol oxidation under low- and high-NO conditions is detected more at the complex ($\sim$12%) than 3-methyl catechol. This suggests that *o*-cresol

oxidation produces an additional isomer or isomers. These isomer(s) are likely less acidic than 3-methyl catechol (i.e, the OH substituents are not on adjacent carbons). To verify this, dihydroxy toluene isomers were tested on the CF$_3$O$^-$ CIMS. 3-methyl catechol and 4-methyl catechol are dominantly detected at m/z (-)143 (0.97 and 0.99, respectively) when considering only the unique fragments (i.e., m/z (-)143, 123, 99, 209, and 189). Conversely, 2-methyl resorcinol and methyl hydroquinone signals are detected at m/z (-) 143, 189, and 209 at the following fractions 0.19, 0.51, & 0.30 and 0.07, 0.29, & 0.64, respectively. As

expected, less acidic isomers of dihydroxy toluene (2-methyl resorcinol and methyl hydroquinone) were detected more at the complex than their more acidic counterparts (3-methyl catechol and 4-methyl catechol).

In the kinetic model (Version 3), dihydroxy, trihydroxy, and tetrahydroxy toluene oxidation products are inferred from the products recommended by MCM v3.3.1 (Jenkin et al., 2003; Bloss et al., 2005) and Olariu et al. (2002) for *o*-cresol oxidation (Figure 3, Section S2). The hydrogen abstraction pathway is assumed to produce 6-methyl-2-nitrophenol and the bicyclic in-

termediate pathway is assumed to produce the unidentified products. Thus, dihydroxy, trihydroxy, and tetrahydroxy toluene are assumed to produce the following: a benzoquinone (0.07), a polyol (0.73), a product from the hydrogen abstraction pathway (0.07), and a product from the bicylic intermediate pathway (0.13). Isomers are not treated separately in the kinetic model. With these additional reactions, when oxidizing *o*-cresol under low-NO conditions, the kinetic model results are similar to the CIMS measurements for first-generation products (dihydroxy toluene) and second-generation products (hydroxy methyl ben-





zoquinone and trihydroxy toluene) (Figure 10a). Although exact yields cannot be reported owing to the lack of availability of authentic standards, yields of ~0.7 for trihydroxy toluene and ~0.1 for hydroxy methyl benzoquinone from dihydroxy toluene oxidation appear to be reasonable. Additionally, the CIMS sensitivity (i.e., the ion-molecule collision rate correction, Section S1) for dihydroxy toluene and trihydroxy toluene is dependent on the isomers that form. The suspected dominant isomers of

dihydroxy toluene (3-methyl catechol) and trihydroxy toluene (2,3,4-trihydroxy toluene) both have a higher sensitivity correction than their other isomer counterparts (Table S1). Thus, dihydroxy toluene and trihydroxytoluene may be underestimated by the CIMS if other isomers form to a large degree.

For 3-methyl catechol oxidation, like *o*-cresol oxidation, OH addition is assumed to occur at the *ortho* position forming 2,3,4-trihydroxy toluene as the dominant isomer. The wall deposition results (Section 4.2.1) imply that multiple isomers of

trihydroxy toluene with different wall loss rates and fragmentation patterns on the CIMS are produced from 3-methyl catechol oxidation. Likely (as theory suggests), 2,3,4-trihydroxy toluene is the dominant isomer, but other isomers also form. Results suggest that the trihydroxy toluene isomers produced during *o*-cresol oxidation versus 3-methyl catechol oxidation are different. The distribution of the complex (m/z (-)225) , F$^-$ transfer (m/z (-)159), and fragment (m/z (-)115) signals for trihydroxy toluene is different during *o*-cresol oxidation (0.38, 0.42, 0.2, respectively) versus 3-methyl catechol oxidation (0.04, 0.42,

0.54, respectively). Likely the distribution of trihydroxy toluene isomers is different between the two cases because 3-methyl catechol is not the sole isomer of dihydroxy toluene produced from *o*-cresol oxidation. The isomers of trihydroxy toluene from *o*-cresol oxidation are detected more at the complex (likely less acidic), while the isomers from 3-methyl catechol oxidation fragment more (likely more acidic).

When 3-methyl catechol is oxidized under low-NO conditions, trihydroxy toluene is over-predicted by the kinetic model

compared to the CIMS measurements (Figure 10c). Likely, the trihydroxy toluene isomers produced from 3-methyl catechol have a lower yield or CIMS sensitivity compared to those produced from *o*-cresol. In order to constrain the exact yield of trihydroxy toluene from dihydroxy toluene, the isomer distribution and wall deposition rate for each isomer need to be understood.

Under high-NO conditions, the kinetic model over-predicts trihydoxy toluene formation compared to the CIMS measurements for both *o*-cresol and 3-methyl catechol oxidation (Figure 10). NO$_3$ oxidation is potentially more significant in the

experiments than the kinetic model predicts because the kinetic model under-predicts methyl nitrophenol, a product from OH and NO$_3$ oxidation of *o*-cresol. Under-predicting the occurrence of NO$_3$ oxidation would lead to an over-prediction of OH oxidation products such as trihydroxy toluene.

Tetrahydroxy toluene and dihydroxy methyl benzoquinone are both over-predicted by the kinetic model as compared to experimental results (Figure 10c). This low yield could be a result of unconstrained loses of tetrahydroxy toluene to the

instrument walls or chamber walls. With the current instrumentation, further understanding of the yield of this product cannot be obtained.

### 4.2.4  Formation of Decomposition Products From Bicyclic Intermediate Pathway

A bicyclic intermediate peroxy radical forms from: 1) OH addition to the ring of an aromatic compound, 2) subsequent O$_2$ addition, 3) cyclization, and 4) another O$_2$ addition. In MCM v3.3.1, this peroxy radical reacts either with NO producing an alkoxy





radical that decomposes or with $HO_2$ producing a hydroperoxide. This hydroperoxide will either react with OH to reform the original $RO_2$ or photolyze to form decomposition products (Figure 3). Here, results suggest that either this hydroperoxide photolyzes more rapidly than MCM v3.3.1 assumes or the initial reaction with $HO_2$ proceeds through two reaction channels: 1) formation of a hydroperoxide and, 2) formation of OH and an alkoxy radical, which rapidly decomposes. Under low-NO

conditions, the kinetic mechanism (Version 1) predicts most of the bicyclic intermediate products are in the form of the bicyclic intermediate hydroperoxide. Conversely, the CIMS measurements imply that most of the bicyclic intermediate products are in the form of decomposition products. $o$-Cresol and 3-methyl catechol oxidation under low- and high-NO conditions produce the same decomposition products rapidly and in high concentration. More specifically, for $o$-cresol oxidation under low-NO conditions, the CIMS detects acetyl acrylic acid, a decomposition product from the bicyclic intermediate pathway, at a peak of

$\sim$1.5 ppb, while the kinetic mechanism predicts much less acetyl acrylic acid forms (peak $\sim$ 0.1 ppb). The $CF_3O^-$ CIMS does not detect the bicyclic intermediate hydroperoxide. Either the signal is below the detection limit or the compound is not stable under $CF_3O^-$ ion chemistry. Birdsall et al. (2010) similarly proposed that some of the bicyclic intermediate peroxy radical from toluene reacts with $HO_2$ to form an alkoxy radial and OH. Additionally, recent studies have identified various peroxy radicals that upon reaction with $HO_2$ do not form a hydroperoxide in unity yield (Orlando and Tyndall, 2012, and references

therein). The additional OH produced from the bicyclic intermediate pathway would help explain why Version 3 of the kinetic model under-predicts the decay of the precursor (Figure 10).

A variety of decomposition products suspected to arise from the bicyclic intermediate pathway were detected by the CIMS (Figures 5 and 6). However, not all products formed from this pathway are detected as the CIMS is not sensitive to unfunctionalized ketones and aldehydes. OH oxidation of $o$-cresol and 3-methyl catechol through the bicyclic intermediate pathway

yields two products, only one of which is typically detected by the CIMS. For trihydroxy toluene and tetrahydroxy toluene, often both products are detected. Many basic simplifications/assumptions are made to compare the CIMS and kinetic model results. All products are estimated to have the same CIMS sensitivity as glycolaldehyde. In the kinetic mechanism (version 3), $o$-cresol, 3-methyl catechol, trihydroxy toluene, and tetrahydroxy toluene are assumed to form a 0.13 yield of bicyclic intermediate products (Section 4.2.3). The OH reaction rate constant for all bicyclic intermediate compounds is assumed to be the

same as that for acetyl acrylic acid (MCM v3.3.1).

With these additional reactions, the measurements of the bicyclic intermediate decomposition products formed from 3-methyl catechol oxidation under low- (Figure S4) and high-NO conditions are well represented by the kinetic model. These same products from $o$-cresol oxidation under low- and high-NO conditions are, however, under-predicted by the kinetic model (but still within uncertainty). Most of the products detected are not unique to one of the precursors (Figures 5 and 6). However,

$C_4H_4O_5$ and $C_5H_6O_5$ are expected to form only from trihydroxy toluene and tetrahydroxy toluene. The CIMS measurements confirm $C_4H_4O_5$ and $C_5H_6O_5$ are produced later in the experiment, as expected for second- or third-generation products (Figure S4).





# 5 Conclusions

OH oxidation of 3-methyl catechol (the dominant OH oxidation product from *o*-cresol) produces the following first-generation products: trihydroxy toluene, hydroxy methyl benzoquinone, and various decomposition products likely from the bicyclic intermediate pathway. Second- and third-generation products from 3-methyl catechol include tetrahydroxy toluene, dihydroxy methyl benzoquinone, pentahydroxy toluene, and trihydroxy methyl benzoquinone. Detection of these products implies that subsequent OH addition to the aromatic ring occurs during *o*-cresol oxidation. Many of these products are expected to be relatively low in volatility (C* $\sim 3.0$ x $10^3$ - 7.7 x $10^{-3}$ $\mu$g m$^{-3}$) and are detected in the particle phase by the DART-MS. Although the gas-phase cresol pathway is relatively minor ($\sim$20%), oxidation products from this pathway are estimated to contribute significantly ($\sim$20-40%) to toluene SOA. Thus, a simple and direct pathway for toluene SOA formation has been identified. Oxidation products from the phenolic pathway of other aromatic compounds are also likely to be important for SOA formation.

*Acknowledgements.* This work was supported by National Science Foundation grants AGS-1240604 and AGS-1523500. We thank Hannah Allen and Anke Noelscher for their experimental assistance and Nathan Dalleska and John Crounse for helpful discussions.

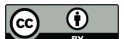



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



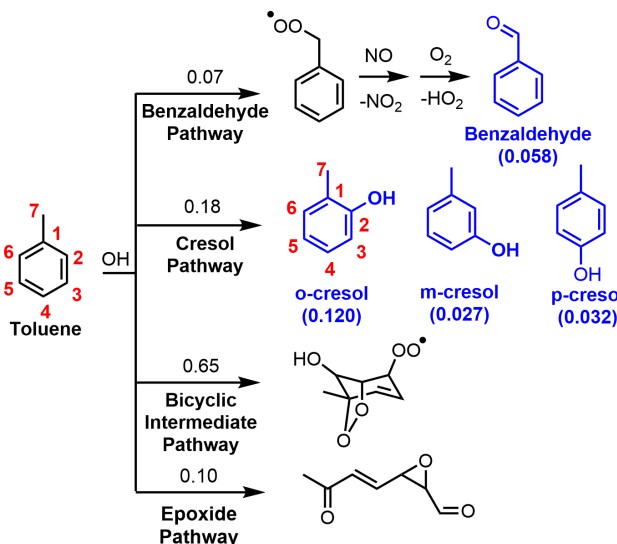

**Figure 1.** Toluene photooxidation pathways from the Master Chemical Mechanism (MCM) v3.3.1 including cresol isomer distribution (Klotz et al., 1998). Ring-retaining products are shown in blue. Carbons on the toluene and *o*-cresol ring structure are labeled in red from 1-7 to facilitate identification of isomers throughout the text.

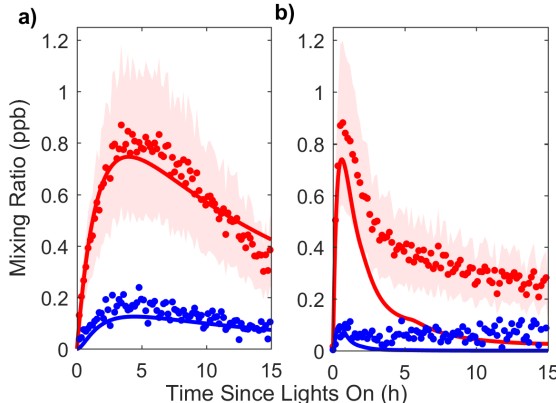

**Figure 2.** Kinetic model predictions (Version 1 solid lines) compared to CIMS measurements (data points) under low-NO (a, experiment 4) and high-NO (b, experiment 3) oxidation of toluene for cresol (red) and dihydroxy toluene (blue). The uncertainty in the CIMS measurements for cresol is shown in red shading. The CIMS measurements under high-NO conditions (panel b) suggest that additional first- or second-generation products that react much slower with OH than cresol give rise to the signal at the same mass.



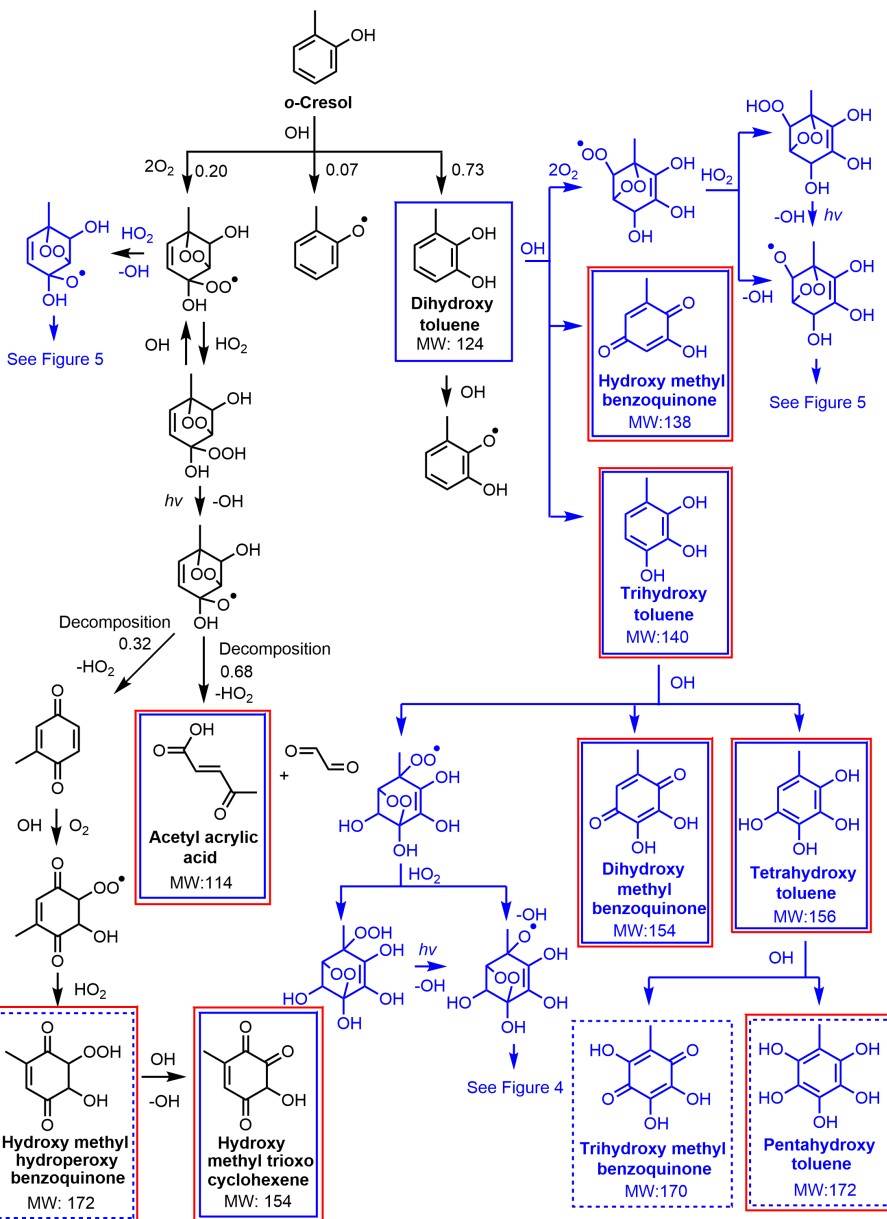

**Figure 3.** Gas-phase chemical mechanism for *o*-cresol photooxidation under low-NO conditions. Recommended pathways by MCM v3.3.1 are shown in black. The proposed mechanism from the present study is shown in blue. Products detected in this study by the CIMS and DART-MS are outlined in blue and red boxes, respectively, with dashed lines indicating compounds detected at only a minimal level.



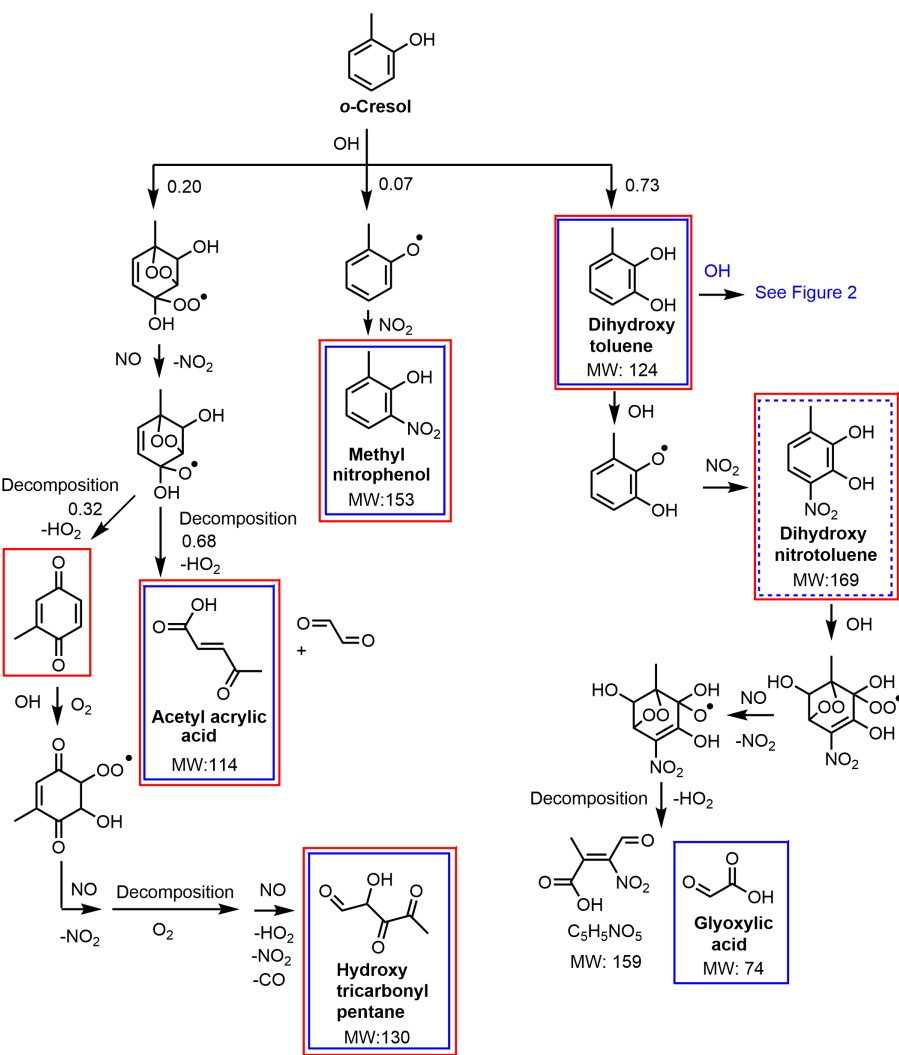

**Figure 4.** Gas-phase chemical mechanism for *o*-cresol photooxidation under high-NO conditions. The MCM v3.3.1 scheme is shown in black. Products outlined in blue were detected in the present study by the CIMS with dashed lines indicating only a minor amount was detected. Red boxed compounds were detected in the present study by the DART-MS.



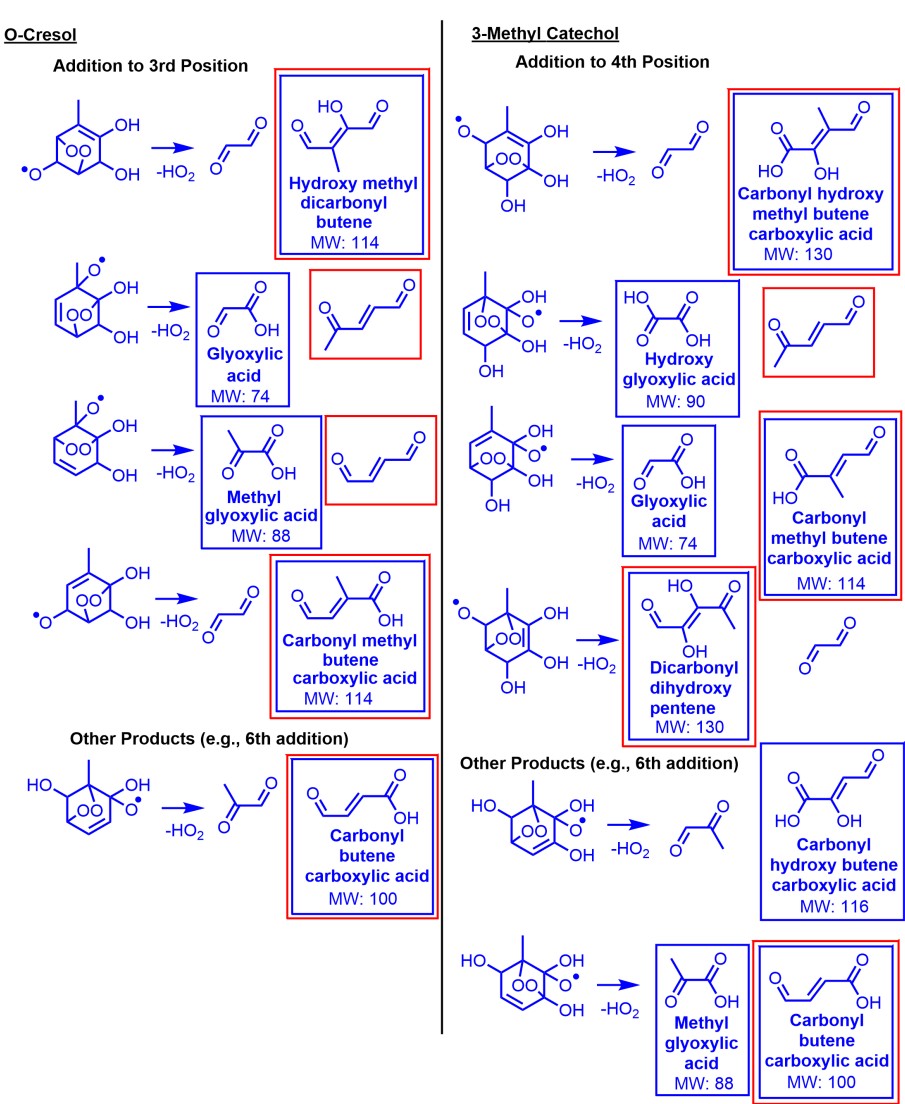

**Figure 5.** Proposed decomposition pathways for bicyclic intermediate compounds formed from OH oxidation of *o*-cresol and 3-methyl catechol. Blue and red boxed compounds were detected by CIMS and DART-MS, respectively.



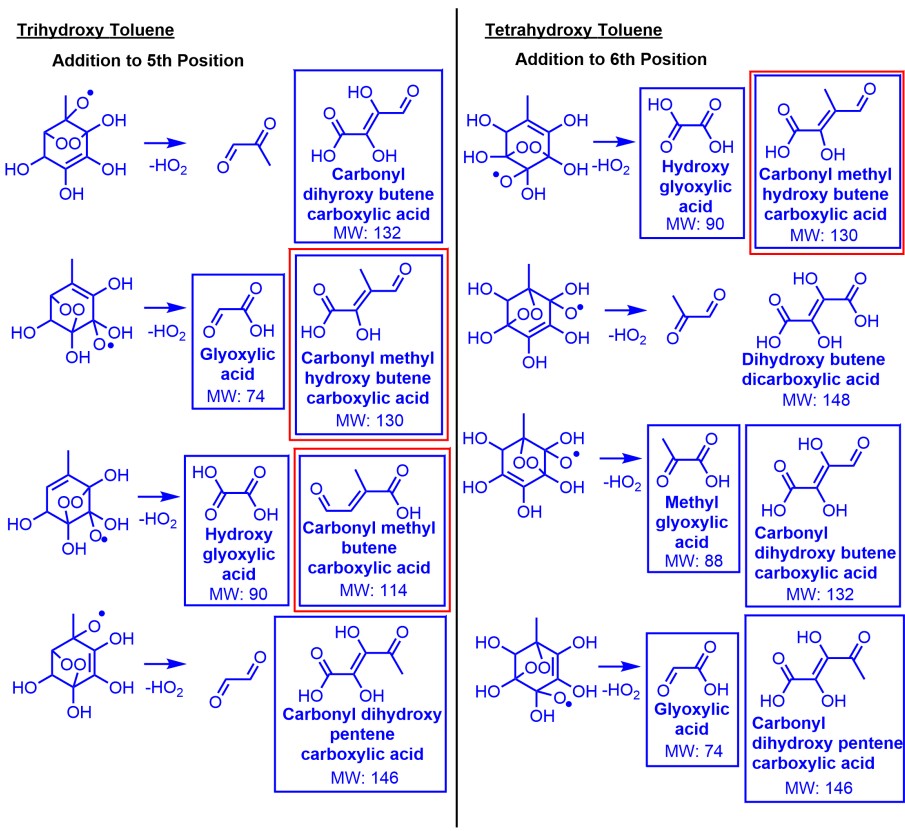

**Figure 6.** Proposed decomposition pathways for bicyclic intermediate compounds formed from OH oxidation of trihydroxytoluene and tetrahydroxytoluene. Blue and red boxed compounds were detected by CIMS and DART-MS, respectively.



**Figure 7.** Gas-phase chemical mechanism for benzaldehyde photooxidation under low- and high-NO conditions. MCM v3.3.1 pathways are shown in black. Products detected by the CIMS are boxed in blue, with dashed lines indicating only a minor amount forms.




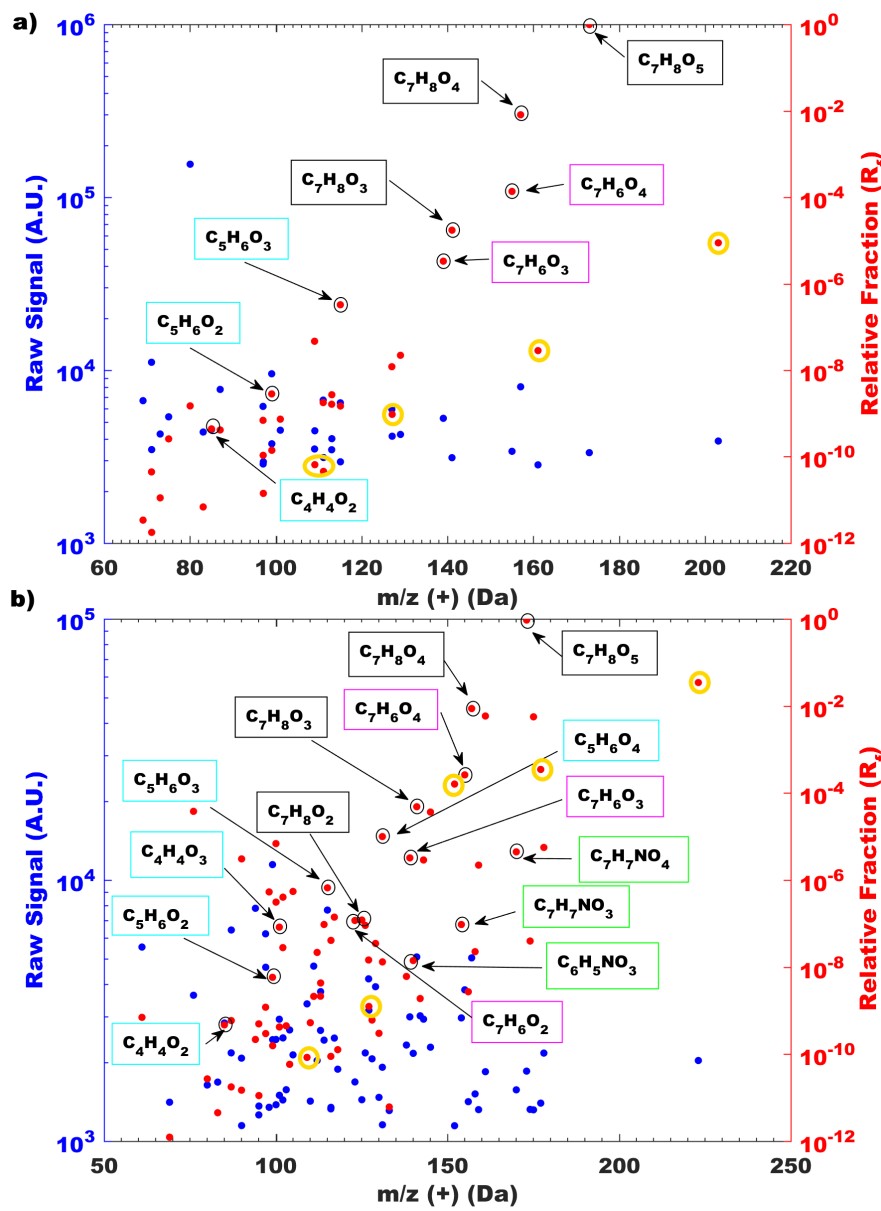

**Figure 8.** Particle-phase products detected by DART-MS during oxidation of toluene under low-NO conditions (a, experiment 13) and high-NO conditions (b, experiment 14) with boxes identifying the following types of compounds: polyols (black), methyl benzoquinone type compounds (magenta), decomposition products from the bicyclic intermediate pathway (cyan), nitro compounds (green), and presumed oligomerization products (i.e., those with >7 carbons, gold).





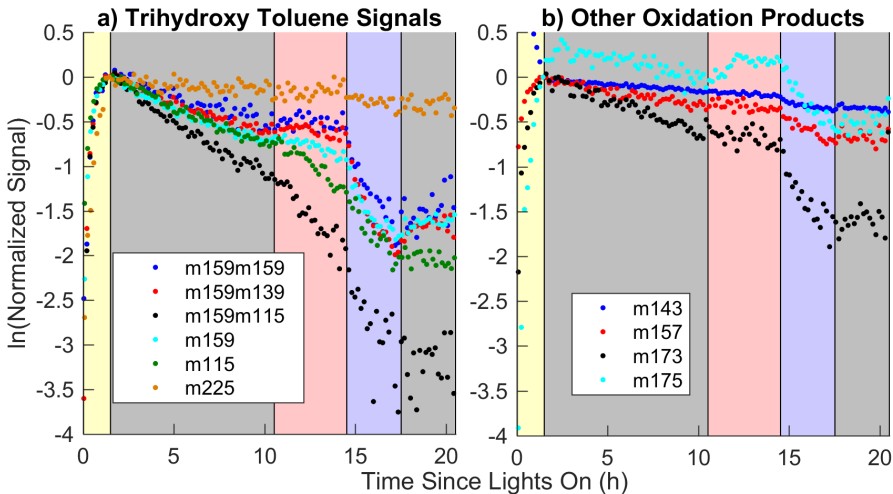

**Figure 9.** CIMS signals for trihydroxy toluene (panel a) and the following compounds (panel b): 3-methyl catechol (blue), hydroxy methyl benzoquinone (red), dihydroxy methyl benzoquinone (black), and tetrahydroxy toluene (cyan) during experiment 9. During this experiment, lights were turned on to generate 3-methyl catechol oxidation products (yellow shading). Then lights were turned off to measure the decay of these products to the chamber walls at 28°C (gray shading). Then the chamber was heated to 42°C (red shading), cooled to 16°C (blue shading), and finally heated back to 28°C (gray shading).





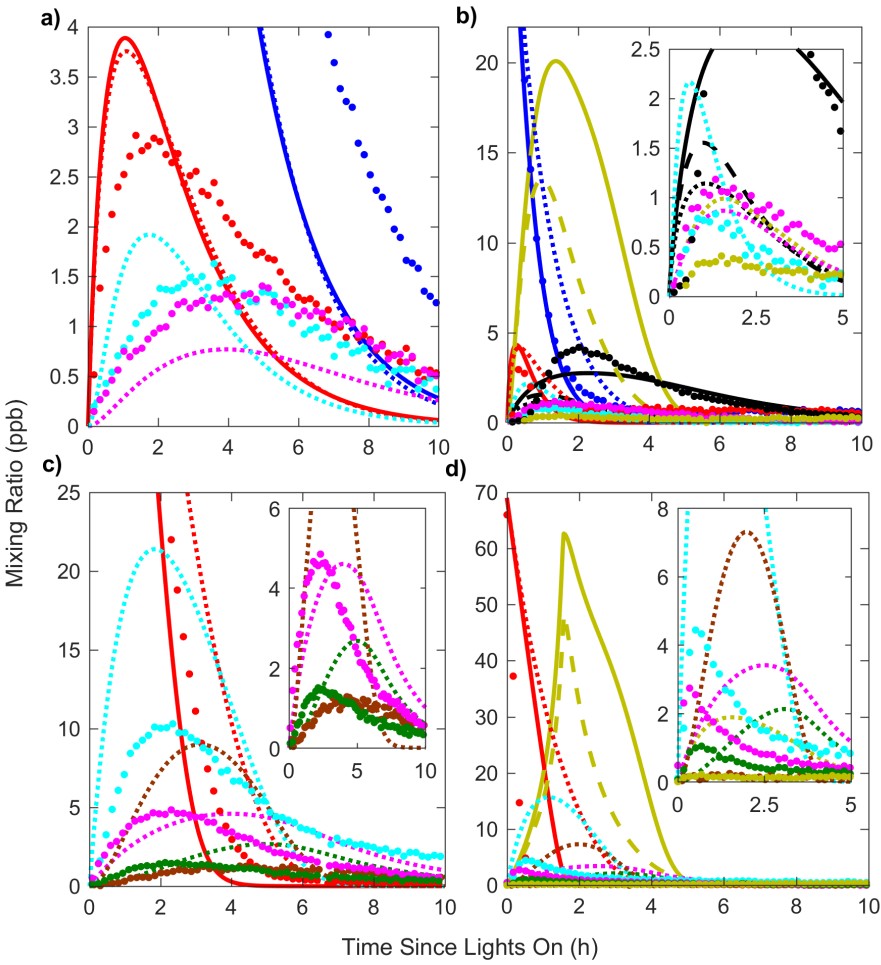

**Figure 10.** Kinetic model predictions (Version 1 solid lines, Version 2 dashed lines, Version 3 dotted lines) versus CIMS measurements (data points) for *o*-cresol oxidation under low-NO (a) and high-NO (b) conditions and 3-methyl catechol oxidation under low-NO (c) and high-NO (d) conditions. When necessary, a finer detail plot of selected compounds with lower signal is shown in the upper right corner. Colors for all graphs are *o*-cresol (blue), 3-methyl catechol (red), trihydroxy toluene (cyan), tetrahydroxy toluene (brown), hydroxy methyl benzoquinone (magenta), dihydroxy methyl benzoquinone (green), hydroxy nitrotoluene (black), and dihydroxy nitrotoluene (gold).



**Table 1.** Description of Experiments

| Expt. # | VOC | VOC (ppb) | Oxidant Precursor | Initial NO (ppb) | Continuous NO Injection (ppb h$^{-1}$) [a] | Temp. (K)/ RH (%) [b] |
|---|---|---|---|---|---|---|
| | | | **Gas-Phase Experiments** [c] | | | |
| 1 | Toluene | 98 | H$_2$O$_2$ + NO | 72 | 98 (149); 61 (223); 30 (UE) | 301 / <3-7 |
| 2 | Toluene | 91 | H$_2$O$_2$ | NA | NA | 301 / <3-5 |
| 3 | Toluene | 49 | H$_2$O$_2$ + NO | 74 | 98 (120); 61 (204); 30 (UE) | 301 / <3-5 |
| 4 | Toluene | 46 | H$_2$O$_2$ | NA | NA | 301 / ≤3 |
| 5 | o-cresol | 40 | H$_2$O$_2$ + NO | 79 | 98 (120); 61 (20); 30 (UE) | 301 / 4-9 |
| 6 | o-cresol | 36 | H$_2$O$_2$ | NA | NA | 301 / <3-4 |
| 7 | 3-methyl catechol | 69 | H$_2$O$_2$ + NO | 74 | 98 (123); 30 (UE) | 301 / 3-7 |
| 8 | 3-methyl catechol | 97 | H$_2$O$_2$ | NA | NA | 301 / ≤3 |
| 9 | 3-methyl catechol | 59 | H$_2$O$_2$ | NA | NA | 289-315/ <14 |
| 10 | 3-methyl catechol | 65 | H$_2$O$_2$ | NA | NA | 302 / <3 |
| 11 | Benzaldehyde | ∼50 | H$_2$O$_2$ | NA | NA | 301 / ≤3 |
| 12 | Benzaldehyde | ∼50 | H$_2$O$_2$ + NO | 72 | 98 (68); 61 (35); 30 (329); 22 (UE) | 301 / ≤3 |
| | | | **Particle-Phase Experiments** [c] | | | |
| 13 | Toluene | 286 | H$_2$O$_2$ | NA | NA | 301 / 3-6 |
| 14 | Toluene | 313 | H$_2$O$_2$ + NO | 78 | 146 (263); 53 (734); 15 (UE) | 300 / <3-7 |
| 15 | o-cresol | 143 | H$_2$O$_2$ + NO | 84 | 146 (262); 53 (182); 15 (UE) | 301 / 9-12 |

[a] NO was continuously injected. The rate of NO injection decreased over the course of the experiment. The following format is used: rate in ppb h$^{-1}$ (number of minutes injected at that rate). "UE" indicates that the rate was used until the end of the experiment. [b] RH = Relative Humidity. [c] Seed aerosol was injected into the chamber for all particle-phase experiments, but not for gas-phase experiments.

**Table 2.** Fraction of CIMS signal detected from the transfer reaction, complex formation, or fragmentation

| | | **Fraction of Signal** | | |
|---|---|---|---|---|
| VOC | Gas | Transfer | Complex | Fragments |
| o-cresol | purified air | 0.12 | 0.77 | 0.11 |
| o-cresol | dry N$_2$ | 0.19 | 0.75 | 0.05 |
| 3-methyl catechol | purified air | 0.78 | 0.02 | 0.20 |
| 3-methyl catechol | dry N$_2$ | 0.90 | <0.01 | 0.10 |



**Table 3.** Estimated vapor pressures and saturation mass concentrations for main products detected by the CIMS from toluene OH oxidation

| VOC | Estimated Vapor Pressure [a] (atm, 298K) | | Saturation Mass Concentration [b] (C*) ($\mu$g m$^{-3}$) | |
|---|---|---|---|---|
| | EVAP | Nann | Evap | Nann |
| Toluene | $3.7 \times 10^{-2}$ [c] | $3.7 \times 10^{-2}$ [c] | $1.4 \times 10^{8}$ | $1.4 \times 10^{8}$ |
| o-cresol | $3.9 \times 10^{-4}$ [c] | $3.9 \times 10^{-4}$ [c] | $1.7 \times 10^{6}$ | $1.7 \times 10^{6}$ |
| Acetyl acrylic acid | $6.6 \times 10^{-6}$ | $1.7 \times 10^{-5}$ | $3.1 \times 10^{4}$ | $7.9 \times 10^{4}$ |
| Hydroxy methyl hydroperoxy benzoquinone | $1.9 \times 10^{-9}$ | U | $1.3 \times 10^{1}$ | NA |
| Hydroxy methyl trioxo cyclohexene | $8.2 \times 10^{-7}$ | U | $5.2 \times 10^{3}$ | NA |
| 3-methyl catechol | $2.3 \times 10^{-6}$ | $6.8 \times 10^{-6}$ | $1.2 \times 10^{4}$ | $3.5 \times 10^{4}$ |
| Hydroxy methyl benzoquinone | $5.3 \times 10^{-7}$ | U | $3.0 \times 10^{3}$ | NA |
| Trihydroxy toluene | $1.1 \times 10^{-8}$ | $6.0 \times 10^{-8}$ | $6.3 \times 10^{1}$ | $3.4 \times 10^{2}$ |
| Dihydroxy methyl benzoquinone | $8.3 \times 10^{-9}$ | U | $5.2 \times 10^{1}$ | NA |
| Tetrahydroxy toluene | $2.4 \times 10^{-11}$ | $3.3 \times 10^{-10}$ | $1.5 \times 10^{-1}$ | 2.1 |
| Trihydroxy methyl benzoquinone | $9.5 \times 10^{-11}$ | U | $6.6 \times 10^{-1}$ | NA |
| Pentahydroxy toluene | $9.0 \times 10^{-14}$ | $1.1 \times 10^{-12}$ | $6.3 \times 10^{-4}$ | $7.7 \times 10^{-3}$ |
| Hydroxy nitrotoluene | U | $1.8 \times 10^{-5}$ | NA | $1.1 \times 10^{5}$ |
| Hydroxy tricarbonyl pentane | $3.1 \times 10^{-4}$ | U | $1.7 \times 10^{6}$ | NA |
| Dihydroxy nitrotoluene | U | $2.0 \times 10^{-7}$ | NA | $1.4 \times 10^{3}$ |
| Glyoxylic acid | $1.9 \times 10^{-3}$ | U | $5.5 \times 10^{5}$ | NA |
| Benzoic acid | $1.1 \times 10^{-5}$ | $8.2 \times 10^{-6}$ | $5.5 \times 10^{4}$ | $4.1 \times 10^{4}$ |
| Peroxybenzoic acid | $8.8 \times 10^{-5}$ | U | $5.0 \times 10^{5}$ | NA |
| Nitrophenol | U | $8.7 \times 10^{-5}$ | NA | $5.0 \times 10^{5}$ |
| Dinitrophenol | U | $3.3 \times 10^{-7}$ | NA | $2.5 \times 10^{3}$ |
| Hydroxy methyl dicarbonyl butene | $1.4 \times 10^{-4}$ | U | $6.5 \times 10^{5}$ | NA |
| Methyl glyoxylic acid | $6.1 \times 10^{-4}$ | U | $2.2 \times 10^{6}$ | NA |
| Carbonyl methyl butene carboxylic acid | $6.6 \times 10^{-6}$ | $1.9 \times 10^{-5}$ | $3.1 \times 10^{4}$ | $8.9 \times 10^{4}$ |
| Carbonyl butene carboxylic acid | $2.0 \times 10^{-5}$ | $3.0 \times 10^{-5}$ | $8.2 \times 10^{4}$ | $1.2 \times 10^{5}$ |
| Carbonyl hydroxy methyl butene carboxylic acid | $1.1 \times 10^{-7}$ | U | $5.9 \times 10^{2}$ | NA |
| Hydroxy glyoxylic acid | $2.9 \times 10^{-7}$ | U | $1.1 \times 10^{3}$ | NA |
| Dicarbonyl dihydroxy pentene | $5.8 \times 10^{-6}$ | U | $3.1 \times 10^{4}$ | NA |
| Carbonyl hydroxy butene carboxylic acid | $3.3 \times 10^{-7}$ | U | $1.6 \times 10^{3}$ | NA |
| Carbonyl dihydroxy butene carboxylic acid | $7.6 \times 10^{-9}$ | U | $4.1 \times 10^{1}$ | NA |
| Carbonyl methyl hydroxy butene carboxylic acid | $1.1 \times 10^{-7}$ | U | $5.9 \times 10^{2}$ | NA |
| Carbonyl dihydroxy pentene carboxylic acid | $2.4 \times 10^{-9}$ | U | $1.4 \times 10^{1}$ | NA |

[a] U = Unable to estimate, EVAP = EVAPORATION and Nann = Nannoolal. [b] Here C* = MW*$P^0$ / ($RT$) where MW = molecular weight, $P^0$ = liquid vapor pressure, $R$ = gas constant, and $T$ = temperature. [c] The values reported are the measured values (Yaws, 1994).