# Peer review of "Formation of Highly Oxygenated Low-Volatility Products from Cresol Oxidation"

_Atmospheric Chemistry and Physics, 2016_

## Referee Comment (RC1) · Anonymous Referee #1 · 7 Nov 2016

General comment:

The authors present a mechanistic study of toluene photo-oxidation. They focus on the chemistry of first and higher generation products form toluene photoxidation. As analytical tool they apply CF3O- CIMS. The results are compared to MCM3.31; and missing parts according to the new results were added /modified in two steps. These model improvements led to better consistency between the model results and measurements. Quantification was in parts inherently limited by absence of suited calibration compounds. Very positively, the authors put some efforts in characterizing the sensitivity of their CIMS for the expected compounds and compound classes. The authors also show the importance of higher generation product to SOA formation. The results are new and interesting, and as toluene is an abundant aromatic VOC, they are an important contribution for understanding VOC degradation and SOA formation in

the atmosphere. The very interesting paper is well structured, and overall well written. However, I had some difficulties to follow some of the parts in the experimental section. I have the impression that this did not depend so much on the level of details that are given, but on notations and "unlucky" formulations. I think with a little effort that could be improved easily. I will list some examples below.

The manuscript should be published in ACP after the authors considered the minor points below.

Minor comments:

The scope of abstract and conclusion are not quite balanced. While the abstract focus more on o-cresol and benzaldehyde, the conclusion focus solely on 3-methyl catechol.

p.1, line 10: this sentence is somehow askew. It requires either reformulation or a reference to the yield 0.7, like "reported yield" or so.

Section 2.1 Experimental Design: here some info about the light source(s) is (are) missing: You refer to H2O2 photolysis as OH source on one hand, but later to jNO2 as a measure of photooxidation and a light source to prevent NO3 formation. I suggest, shortly to describe the main features / spectral dependence of your light source.

p.5, line 1f: this is difficult to follow. Why do you speak about complex interaction when you obviously address the analyte*CF3O complex. Similar the analyte*F- adduct is formed by F transfer, but formation process and result are not identical. Also p.5, line 25 ff: "Traditionally, an analyte (A) is detected either at the F− transfer reaction (A+19) or complex formation (A+85)." Probably, better "detected" as "F- adduct" or "CF3O-complex"?!

p.5, line 8: is the 500 ml glass bulb the FTIR cell ?

p.5, line 14ff: I don't understand what are result and consequence from these comparisons. Please, clarify.

p.5, line 20: I guess the bulb was meanwhile empty of o-cresol?

p.5, line 28: it would be helpful to mention that "your" purified air is somewhat humid (RH?). Or is it O2 in air vs N2 that makes the difference?

p.5, line28: "Likely the presence of water destabilizes the molecular ion formed from CF3O− ionization leading to more fragmentation." I guess, you mean that the presence of water affects F- transfer adduct most?! Please clarify.

p.6, line4ff: I find the use of the word "consistent" difficult to misleading (here and at some instances). Please, check and maybe reformulate to be clearer.

p.6, line8: I guess you mean that a calibration with purified air at RH??, lead to the same results as dosing water to the system? Please, clarify.

p.6, line20: "Unlike complex interactions, F− transfer reactions are increasingly likely to decompose into smaller fragments". Compare my comments above; I suggest to clearly separate the notation for the "formation process" and the "resulting ion"

p.6, line22: "Unlike o-cresol, the sum of all unique signals and small fragmentation products for3-methyl catechol is not consistent for the relative humidities used in these experiments.", Again what does "consistent" mean here?

p.6, line26: I guess that is because the RH increases in the course of the experiments. Please, clarify.

p.7, line14: What is an array of signals?

p.10, line18: in Figure 3, you use notation "dihydroxy toluene". Please make clearer that this is the same as 3-methyl catechol or replace it.

p.10, line 21: the info that this referring to the high NOX experiment and Figure 4 is missing.

p.13, line 28: Shouldn't the yield be dependent on the organic mass produced? Please,

discuss more. p.14, line11: typo "theoretical"

p.14, line22: maybe better "increasing OH substitution"?

p.14, line 26: this sentence is difficult to follow, please reformulate.

p.14, line 29: "favors" the O2 reactions channel, maybe better.

p.17, line 20: typo, These isomers. . .

p.18, line 24: I understood you destroyed NO3 by visible light (p3, line 25)?

p.19, line 17: do you mean "expected"?

p.19, line 19: I cannot relate these statements to what is shown in Figure 5 and 6. There are more than two, i.e. many products generated via bicyclic pathway.

p.19, line 27: I don't understand Figure S4 and its caption. What is matching and not matching in this Figure?

Suppl. p.7, line1: if it is just the branching, the modelled sum should be ok. It looks like it, does it?!

Suppl. p.7, line 9: typo "Nitrosophenol"
* * *

---

## Referee Comment (RC2) · Anonymous Referee #2 · 1 Dec 2016

General This is a very interesting and impressive paper on the oxidation of cresol studied at the Caltech chamber. Generally, it contains a wealth of new findings extending the knowledge on aromatics oxidation considerably.

Apparently, despite the study of aromatics oxidation over some decades now, this appears another field where the now available mass spectometric techniques allow for the identification of reaction products which have not been identified before, especially for low NO conditions - but see and consider the note below.

On the other hand, the paper confirms a lot of findings as they are implemented tinto the MCM 3.3.1, many of them for high NO conditions which are expected to be met in regions where aromatic VOC emissions coincide with elevated NO and NOx levels.

It seems that especially for low NO conditions many new observations are made but

the authors should include into their paper a discussion where conditions with low NO enabling prefence of ROO + HO2 over ROO + NO can really be met.

The paper has the potential to deliver valuable changes and additions to chemical schemes such as the MCM 3.3.1. as it conains infomation of the formation of products in oxidation generations beyond one and two.

Overall, I think the paper is a great step forward in the understanding of aromatics oxidation and merits publication in ACP subject to only minor revision.

Details

p3, line 27: While it is understandable to avoid further complications during the present study, it would be extremely interesting to do a similar study at higher RH and then see the coupling to aqueous chemistry

p18, section 4.2.4: Maybe a reference can be given for the classic formation of the endo-peroxide ? I wonder if a intramolecular H-shift could occur in systems like the peroxyl / endoperoxides at the different stages of the mechanisms discussed so that a hydroperoxide might form from this and not only by + HO2 / - O2 ? Maybe it would be timeöy to discuss such possibility.

Figure 3: The main additions here are on the catechol oxidation.

Figure 4: I do not understand "See Figure 2 " written at the catechol up right in the scheme. Probabyl this should be 'Figure 3 ' ? Please explain / correct.

References: I wonder if the paper

PAN, Shan-Shan, and Li-Ming WANG. "The Atmospheric Oxidation Mechanism of o-Xylene Initiated by Hydroxyl Radicals." Acta Physico-Chimica Sinica 31.12 (2015): 2259-2268.

should be mentioned because it contains remarkable recent considerations on xylene oxidation

---

## Author Comment (AC1) · 13 Jan 2017

Response to Review 1

Comment: "The authors present a mechanistic study of toluene photooxidation. They focus on the chemistry of first and higher generation products form toluene photoxidation. As analytical tool they apply CF3O- CIMS. The results are compared to MCM3.31; and missing parts according to the new results were added /modified in two steps. These model improvements led to better consistency between the model results and measurements. Quantification was in parts inherently limited by absence of suited calibration compounds. Very positively, the authors put some efforts in characterizing the sensitivity of their CIMS for the expected compounds and compound classes. The authors also show the importance of higher generation product to SOA formation. The

results are new and interesting, and as toluene is an abundant aromatic VOC, they are an important contribution for understanding VOC degradation and SOA formation in the atmosphere. The very interesting paper is well structured, and overall well written. However, I had some difficulties to follow some of the parts in the experimental section. I have the impression that this did not depend so much on the level of details that are given, but on notations and "unlucky" formulations. I think with a little effort that could be improved easily. I will list some examples below."

Response: Thank you for the helpful comments and suggestions. As suggested, we have edited the experimental section to make it clearer. We have attached as a supplement a pdf file including a comparison between the ACPD version and the edited version for ease of viewing changes.

——

Comment: "The scope of abstract and conclusion are not quite balanced. While the abstract focus more on o-cresol and benzaldehyde, the conclusion focus solely on 3-methyl catechol."

Response: We have revised the conclusion to better balance the focus.

——

Comment: "p.1, line 10: this sentence is somehow askew. It requires either reformulation or a reference to the yield 0.7, like "reported yield" or so."

Response: We have added "reported yield" and included a reference.

——

Comment: "Section 2.1 Experimental Design: here some info about the light source(s) is (are) missing: You refer to H2O2 photolysis as OH source on one hand, but later to jNO2 as a measure of photooxidation and a light source to prevent NO3 formation. I suggest, shortly to describe the main features / spectral dependence of your light

source."

Response: Thanks for this observation. We have added more description about the light source used in the Caltech chamber. Also we listed more photolysis rate constants and provided more description about how NO3 photolysis impacts the results in this study.

——

Comment: "p.5, line 1f: this is difficult to follow. Why do you speak about complex interaction when you obviously address the analyte*CF3O complex. Similar the analyte*F-adduct is formed by F transfer, but formation process and result are not identical. Also p.5, line 25 ff: "Traditionally, an analyte (A) is detected either at the F− transfer reaction (A+19) or complex formation (A+85)." Probably, better "detected" as "F- adduct" or "CF3O- complex"?!"

Response: Thanks for this observation. We have restructured this section and also revised the terminology we are using to make this clearer. As recommended, we revised the labeling to separate the terminology used for the formation process and the resulting ion. We also added in more reference to the reactions (R1-R6) to further define and remind the reader of the terminology used.

——

Comment: "p.5, line 8: is the 500 ml glass bulb the FTIR cell ?"

Response: We apologize for the confusion. It is a separate glass bulb. We have revised this section to make this clearer.

——

Comment: "p.5, line 14ff: I don't understand what are result and consequence from these comparisons. Please, clarify."

Response: Thanks for noticing this. We have revised this paragraph and the following

paragraph to make the results and consequences for these comparisons clearer. In summary: There is only one study (Etzkorn et al. 1999) quantifying o-cresol with FT-IR. o-cresol is difficult to quantify because it is easily lost to surfaces, so we wanted to independently verify the Etzkorn et al. 1999 results. Since there are no similar studies for o-cresol, we compared the m-cresol results from Etzkorn et al., 1999 to those from PNNL.

——

Comment: "p.5, line 20: I guess the bulb was meanwhile empty of o-cresol?"

Response: We have revised the text to be clearer. In summary: The FT-IR cell was filled with o-cresol. Then over more than an hour sequential FT-IR measurements were performed to monitor the loss of o-cresol to the glass cell walls. ——

Comment: "p.5, line 28: it would be helpful to mention that "your" purified air is somewhat humid (RH?). Or is it O2 in air vs N2 that makes the difference?"

Response: We have removed this sentence and instead only referenced Table 2. We have also revised Table 2 and added in a note describing the approximate RH levels in the dry N2 versus purified air.

——

Comment: "p.5, line28: "Likely the presence of water destabilizes the molecular ion formed from CF3O− ionization leading to more fragmentation." I guess, you mean that the presence of water affects F- transfer adduct most?! Please clarify."

Response: We have revised this idea throughout the methods section to make this clearer. We have also decided to explain the F- transfer adduct as a fragment itself.

——

Comment: "p.6, line4ff: I find the use of the word "consistent" difficult to misleading (here and at some instances). Please, check and maybe reformulate to be clearer."

Response: We have changed this instance of "consistant" to "stable" to avoid confusion. We also updated the use of "not consistant" when describing the 3-methyl catechol water calibration to "decreased".

——

Comment: "p.6, line8: I guess you mean that a calibration with purified air at RH??, lead to the same results as dosing water to the system? Please, clarify."

Response: We apologize for the confusion. Your assumption is right; we have revised this sentence to make this clearer.

——

Comment: "p.6, line20: "Unlike complex interactions, F$-$ transfer reactions are increasingly likely to decompose into smaller fragments". Compare my comments above; I suggest to clearly separate the notation for the "formation process" and the "resulting ion""

Response: We have fixed this terminology throughout this section to be clear when we are describing the process of ionization versus when we are referring to an actual ion. Throughout the text to improve clarity, we now state "ion" whenever we are referring to the ion itself.

——

Comment: "p.6, line22: "Unlike o-cresol, the sum of all unique signals and small fragmentation products for3-methyl catechol is not consistent for the relative humidities used in these experiments.", Again what does "consistent" mean here?"

Response: Here we changed "consistent" to "decreased as the relative humidity increased". Also we revised the next sentence to further explain why this decrease occurred.

——

Comment: "p.6, line26: I guess that is because the RH increases in the course of the experiments. Please, clarify."

Response: We have changed "change" to "increase", to be more explicit. The RH increases slightly over the course of the experiments because of minor leaks in the Teflon chamber walls.

——

Comment: "p.7, line14: What is an array of signals?"

Response: We have revised this sentence for clarity. We meant that there was a wide variety of other signals as well as those mentioned in the previous paragraph.

——

Comment: "p.10, line18: in Figure 3, you use notation "dihydroxy toluene". Please make clearer that this is the same as 3-methyl catechol or replace it."

Response: We apologize for the confusion. 3-methyl catechol is the dominant isomer of dihydroxy toluene. Our CIMS cannot differentiate between the isomers of dihydroxy toluene, so throughout most of the text we refer to "dihydroxy toluene" as dihydroxy toluene since the isomers are not known. We refer to 3-methyl catechol when we inject it into the chamber because then the exact isomer is known. We have revised the description in figure 3 to clarify this and also added in more description in the second paragraph of section 3.1.

——

Comment: "p.10, line 21: the info that this referring to the high NOX experiment and Figure 4 is missing."

Response: We added in "under high-NO conditions" and referenced Figures 4 and 10b&d as suggested.

——

Comment: "p.13, line 28: Shouldn't the yield be dependent on the organic mass produced? Please, discuss more. "

Response: We have added in several sentences to better explain the method we choose to estimate the contribution of cresol to toluene SOA. Mainly, the experiments in this study were specifically designed to investigate chemistry rather than SOA yields. SOA yield studies are performed differently than the studies used in this work. For example, most experiments in this study were performed without seed aerosol. Instead we use the SOA yields measured by Zhang et al., 2014; These experiments were specifically designed to measure SOA yields and take into account particle and vapor wall losses.

——

Comment: "p.14, line11: typo "theoretical""

Response: Thanks, this has been corrected.

——

Comment: "p.14, line22: maybe better "increasing OH substitution"?"

Response: Thanks, this has been revised as suggested.

——

Comment: "p.14, line 26: this sentence is difficult to follow, please reformulate."

Response: This sentence has been reformulated.

——

Comment: "p.14, line 29: "favors" the O2 reactions channel, maybe better."

Response: We have restructured this sentence.

Comment: "p.17, line 20: typo, These isomers. . ."

Response: We left this as isomer(s) to represent that this could be one isomer or several isomers. We cannot be certain with the present technique how many different isomers are present.

Comment: "p.18, line 24: I understood you destroyed NO3 by visible light (p3, line 25)?"

Response: We apologize for the confusion. We have revised this paragraph to better explain our reasoning. Also we have added in the methods section the rate constant for NO3 photolysis and explained the type of lights used.

Comment: "p.19, line 17: do you mean "expected"?"

Response: We changed "suspected" to "assumed". Our point in using "assumed" is that we only know that various decomposition products form, and we "assume" or "suspect" they are from the bicyclic intermediate pathway, but this is based on other studies. Unfortunately, our CIMS is not sensitive to the bicyclic intermediate product.

Comment: "p.19, line 19: I cannot relate these statements to what is shown in Figure 5 and 6. There are more than two, i.e. many products generated via bicyclic pathway."

Response: We apologize for the confusion. We have revised this terminology to state that two product types form: a functionalized ketone/aldehyde and an unfunctionalized ketone/aldehyde.

Comment: "p.19, line 27: I don't understand Figure S4 and its caption. What is matching and not matching in this Figure?"

Response: We have added another couple sentences in the supplement to describe this figure. See last paragraph of Section S1.

——

Comment: "Suppl. p.7, line1: if it is just the branching, the modelled sum should be ok. It looks like it, does it?!"

Response: For clarity, we added a reminder here that phenyl hydroperoxide was not detected by the CIMS either because it does not form or is not stable in the ion chemistry of the CF3O- CIMS as stated in the main text. If the phenyl hydroperoxide forms and is not detected by the CIMS then yes you are right the sum of peroxybenzoic acid and benzoic acid for both the experimental data and kinetic model results are fairly similar. However, if phenyl hydroperoxide does not form, then the kinetic model predicts more products than are detected by the CIMS.

——

Comment: "Suppl. p.7, line 9: typo "Nitrosophenol""

Response: Thanks, this has been corrected.

Please also note the supplement to this comment:
http://www.atmos-chem-phys-discuss.net/acp-2016-887/acp-2016-887-AC1-supplement.pdf

**Supplement:**

[revised manuscript text omitted]

**S1 Further Details on $CF_3O^-$ CIMS Analysis**

[Figure]

**Figure S1.** CIMS MS signals of 3-methyl catechol oxidation products (panel a) and MS/MS signals of tetrahydroxy toluene (panel b) for experiment 10. Desorption of compounds from instrument walls was measured by sampling photooxidation products generated in the chamber (yellow) and then immediately switching to purified air (white). *CIMS signal is normalized to time right before lights off.

**Table S1:** Estimated CIMS sensitivity factors

| Compound | Structure | Polarizability ($\overset{\circ}{A}{}^3$) [a] | Dipole Moment (D) [b] | Sensitivity Factor [c] | Notes |
|---|---|---|---|---|---|
| Toluene Related Compounds | | | | | |
| *o*-Cresol | | 11.8 | 1.42 | 1 | |

**Table S1:** Estimated CIMS sensitivity factors

| Compound | Structure | Polarizability ($\overset{\circ}{A}{}^{3}$) [a] | Dipole Moment (D) [b] | Sensitivity Factor [c] | Notes |
|---|---|---|---|---|---|
| *m*-Cresol | | 13.1 | 1.53 | 1.07 | |
| *p*-Cresol | | 13 | 1.53 | 1.06 | |
| 4-Methylcatechol | | 13.7 | 2.7 | 1.44 | |
| Methyl hydro-quinone | | 13.7 | 2.05 | 1.21 | Assume same polarizability as 4-methyl catechol |
| 3-Methylcatechol | | 13.7 | 2.42 | 1.34 | Assume same polarizability as 4-methyl catechol |
| 2-Methyl resorci-nol | | 13.7 | 2 | 1.19 | Assume same polarizability as 4-methyl catechol |
| 4-Methyl resorci-nol | | 13.7 | 1.81 | 1.13 | Assume same polarizability as 4-methyl catechol |
| 5-Methyl resorci-nol | | 13.7 | 2.1 | 1.23 | Assume same polarizability as 4-methyl catechol |
| 4-Methyl-2-nitrophenol | | 16.2 | 3.49 | 1.69 | |

**Table S1:** Estimated CIMS sensitivity factors

| Compound | Structure | Polarizability ($\overset{\circ}{A}{}^3$) [a] | Dipole Moment (D) [b] | Sensitivity Factor [c] | Notes |
|---|---|---|---|---|---|
| Benzoic acid | | 11.3 | 1.26 | 0.92 | |
| | | Benzene Related Compounds | | | |
| Phenol | | 11.1 | 1.54 | 1 | |
| Catechol | | 13.1 | 2.64 | 1.37 | |
| Hydroquinone | | 13.1 | 1.78 | 1.08 | Assume polarizability same as catechol. |
| Resorcinol | | 13.1 | 2.04 | 1.16 | Assume polarizability same as catechol. |
| 1,2,3-Benzene triol | | 11.1 | 3.17 | 1.47 | |
| 1,3,5-Trihydroxy benzene | | 11.1 | 2.7 | 1.32 | Assume polarizability same as 1,2,3-benzene triol |
| o-Nitrophenol | | 14 | 3.12 | 1.48 | |
| m-Nitrophenol | | 14 | 3.89 | 1.73 | Assume polarizability same as o-nitrophenol |

**Table S1:** Estimated CIMS sensitivity factors

| Compound | Structure | Polarizability ($\mathring{A}^3$) [a] | Dipole Moment (D) [b] | Sensitivity Factor [c] | Notes |
|---|---|---|---|---|---|
| p-Nitrophenol | | 14 | 4.9 | 2.06 | Assume polarizability same as o-nitrophenol |
| Nitrohydroquinone | | 14 | 3.5 | 1.60 | Assume polarizability same as o-nitrophenol |
| 3-Nitrocatechol | | 16.5 | 2.1 | 1.16 | Assume polarity increases by same factor as phenol to catechol |
| 4-Nitrocatechol | | 16.5 | 4.95 | 2.07 | Assume polarity increases by same factor as phenol to catechol |
| 2-Nitroresorcinol | | 16.5 | 2.18 | 1.19 | Assume polarity increases by same factor as phenol to catechol |
| 4-Nitroresorcinol | | 16.5 | 4.44 | 1.91 | Assume polarity increases by same factor as phenol to catechol |
| 5-nitroresorcinol | | 16.5 | 3.9 | 1.74 | Assume polarity increases by same factor as phenol to catechol |

[a] Polarizability was estimated using the refractive index of each compound reported in Lide (2001) as done by Dewar and Stewart (1984). [b] The reported dipole moment is the average of all values reported in McClellan (1974) for experiments using benzene as a solvent and taken between 20-30 °C. [c] The sensitivity factor equals the ion-molecule collision rate of the compound divided by the ion-molecule collision rate of o-cresol for toluene related compounds or phenol for benzene related compounds.

As done by Dewar and Stewart (1984), polarizability was estimated using the refractive index reported in Lide (2001) and the formula: $\bar{P} = (3/4\pi N)(M/d)[(n^2-1)/(n^2+2)] * 10^{24}$ where $\bar{P}$ is the average polarizability, n is the refractive index, N

**Table S2.** Water curve correction and sensitivity factors applied to each compound of interest

| Compound | Water Curve Correction | Compound on which Sensitivity Factor is Based [a] |
|---|---|---|
| Cresol | *o*-cresol | Weighted *o*-,*m*-, and *p*-cresol [b] |
| Dihydroxy toluene | 3-methyl catechol | 3-methyl catechol |
| Trihydroxy toluene | 3-methyl catechol | 1,2,3 benzene triol |
| Tetrahydroxy toluene | 3-methyl catechol | 1,2,3 benzene triol |
| Hydroxy methyl benzoquinone | *o*-cresol | *o*-cresol |
| Dihydroxy methyl benzoquinone | 3-methyl catechol | 3-methyl catechol |
| Methyl nitrophenol | o-cresol | 4-methyl-2-nitrophenol |
| Dihydroxy nitrotoluene | 3-methyl catechol | 3-nitrocatechol |
| Benzoic acid | *o*-cresol | benzoic acid |
| Peroxy benzoic acid | *o*-cresol | benzoic acid |
| Phenyl hydroperoxide | *o*-cresol | benzoic acid |
| Nitrosophenol | *o*-cresol | *o*-nitrophenol |
| Nitrophenol | *o*-cresol | *o*-nitrophenol |
| Dinitrophenol | *o*-cresol | *o*-nitrophenol |

[a] The sensitivity factors are listed in Table S1. [b] The photooxidation isomer distribution reported by Klotz et al. (1998) was used to create a generalized cresol sensitivity factor.

is Avogadro's number, M is the molecular weight, and d is the density. The dipole moments measured in benzene and reported by McClellan (1974) were used to estimate the CIMS sensitivity. Dipole moments measured in air would be more accurate than those measured in benzene. However, very few dipole moments measured in air are available for the aromatic compounds of interest. For phenol, the CIMS sensitivity decreases by 7% when using the dipole moment measured in air (Pedersen et al.,

5   1969) versus benzene (McClellan, 1974).

As noted in Table S1 when refractive index was unavailable, the polarizability for the closest related compound was used. The ion-molecule collision rate for each compound was estimated using the polarizabilities and dipole moments reported in Table S1 and the technique described in Su and Chesnavich (1982). The sensitivity is expected to be proportional to the ion-molecule collision rate. The sensitivity factor reported in Table S1 is the ratio of the ion-molecule collision rate for the

10  compound to that of *o*-cresol for toluene related compounds and phenol for benzene related compounds.

As stated in the main text, the *o*-cresol or 3-methyl catechol water curve was used to determine the sensitivity of a compound with a correction for the ion-molecule collision rate. In Table S2, the water curve correction and the sensitivity factor used for each compound is reported. In some cases, as specified in Table S2 the polarizability and dipole moments were not available for toluene related compounds, so the benzene counterpart was used instead. Note that depending on the fraction of isomers

15  of dihydroxy toluene that form from *o*-cresol oxidation, dihydroxy toluene may be underestimated. 3-methyl catechol has the highest sensitivity of all the isomers that could form from *o*-cresol oxidation (3-methyl catechol, 2-methyl resorcinol, 4-

methyl resorcionol, and methyl hydroquinone). Similarly, depending on the exact isomer distribution that forms from dihydroxy toluene oxidation, trihydroxy toluene may be underestimated. 1,3,5-trihydroxy benzene has a lower sensitivity factor (1.32) compared to that for 1,2,3 benzene triol (1.47). Polarizability and dipole moment measurements are not available for hydroxy methyl benzoquinone or dihydroxy methyl benzoquinone. Thus, we assume that hydroxy methyl benzoquinone behaves like

5    *o*-cresol and dihydroxy methyl benzoquinone behaves like 3-methyl catechol.

[Figure]

**Figure S2.** CIMS measurements (data points) compared to predictions from version 1 of kinetic model (lines) for benzaldehyde low-NO oxidation (experiment 10) for the following compounds benzaldehyde (black), peroxybenzoic acid (blue), benzoic acid (red), and phenyl hydroperoxide (cyan).

CIMS measurements and kinetic model results for products from low-NO oxidation of benzaldehyde are displayed in Figure S2. As stated in the main text, phenyl hydroperoxide is not detected by the CIMS either because it does not form or is not stable under the ion chemistry of the $CF_3O^-$ CIMS. Benzoic acid is under-predicted by the kinetic model suggesting it is formed in a higher yield from $RO_2 + RO_2$ reactions, $RO_2 + HO_2$ reactions, or both. The low yield measured by the CIMS of

10   peroxybenzoic acid, a product from only $RO_2 + HO_2$ reaction, could be caused by a variety of factors. For example, if the $RO_2 + RO_2$ reaction rate constant used in MCM v3.3.1 is too low, more $RO_2 + HO_2$ reactions would occur in the kinetic model producing an over-prediction of peroxybenzoic acid. Another possibility is that the branching ratio for the $RO_2 + HO_2$ reaction favors formation of benzoic acid more so than recommended by MCM v3.3.1. Because benzoic acid is a product from both $RO_2 + RO_2$ and $RO_2 + HO_2$ reactions further constraint is not possible.

[Figure]

**Figure S3.** CIMS measurements (data points) compared to predictions from the kinetic model (solid lines version 1 and dotted lines version 3) for benzaldehyde oxidation under high-NO conditions (experiment 11) for the following compounds benzaldehyde (black), nitrophenol (blue), nitrosophenol (cyan), dinitrophenol (red), and maleic anhydride (magenta).

Nitrosophenol is detected from benzaldehyde oxidation under high-NO conditions (Figure S3). Previous studies have detected a product ($C_6H_5O(NO)$) from the reaction of phenoxy with NO (Tao and Li, 1999). The exact isomer that forms has not been experimentally confirmed. Based on theory, nitrosophenol is the most stable isomer (Yu et al., 1995). Two kinetic studies (Berho et al., 1998; Yu et al., 1995) proposed that phenyl nitrite is the dominant isomer given that nitrosophenol, which requires rearrangement, would not form at the timescales of their studies. $C_6H_5O(NO)$ was detected at the fluorine transfer at m/z (-) 142, implying that it is acidic like nitrosophenol. Possibly, nitrosophenol is over-predicted by version 3 of the kinetic model (Figure S3) because two isomers (nitrosophenol and phenyl nitrite) form and the CIMS is only sensitive to nitrosophenol. The reaction rate constant for $C_6H_5O + NO$ measured by Berho et al. (1998) (1.65 x $10^{-12}$ cm$^3$ molec$^{-1}$ s$^{-1}$) is used in the revised mechanism. The reaction of $C_6H_5O + NO$ has been shown to be reversible, but not at temperatures relevant to this study (Berho et al., 1998; Yu et al., 1995).

m/z (-) 183, assumed to be a fragment of dinitrophenol, is possibly also maleic anhydride (cluster). Maleic anhydride is a decomposition product from dinitrophenol in MCM v3.3.1. However, the predicted amount of maleic anhydride formed in the kinetic mechanism (version 1 and 3) is ∼0.2 ppb after 18 hours of oxidation (Figure S3). Additionally, all nitro products detected by the CIMS have a corresponding fragment at the the F$^-$ transfer minus 20 (hydroxy nitrotoluene, dihydroxy nitrotoluene, and nitrophenol). Thus, the m/z (-) 183 signal is attributed to dinitrophenol.

[Figure]

**Figure S4.** CIMS measurements (data points) compared to predictions from version 3 of the kinetic model (lines) for 3-methyl catechol oxidation under low-NO conditions for bicyclic intermediate products from all precursors (black), 3-methyl catechol (blue), trihydroxy toluene (red), tetrahydroxy toluene (magenta), and trihydroxy toluene or tetrahydroxy toluene tracers (cyan).

Figure S4 compares CIMS measurements and kinetic model results for the bicyclic intermediate products. The sum of all bicyclic intermediate products detected by the CIMS and predicted by the kinetic model are shown in black. Given the large approximations outlined in Section 4.2.4 of the main text, the CIMS and kinetic model results are fairly consistent. Also CIMS measurements indicate that bicyclic intermediate products produced from later generation compounds such as trihydroxy
5   toluene and tetrahydroxy toluene (cyan) peak later in the experiment as expected.

**S2    Further Details on Kinetic Model**

The initial conditions specified in Table 1 of the main text were used as input in the kinetic model. The kinetic model was run with 3 different versions. Version 1, the base case of the kinetic model, included reactions from MCM v3.3.1 for toluene and inorganic gas-phase chemistry and experimentally derived wall loss rates of *o*-cresol and 3-methyl catechol. Version 2 includes
10  all reactions in Version 1 and photolysis of hydroxy nitrotoluene and dihydroxy nitrotoluene. Version 3 includes all reactions in Version 2 and oxidation products for 3-methyl catechol and benzaldehyde. The reactions and rate constants are listed in Table S3 and abbreviations are defined in Table S4. These reactions were included to test the chemistry proposed in the main text. Exact branching ratios and reaction rates for these reactions are unknown. Estimates based on known reactions of similar compounds were used.
15  Hydrogen abstraction from the hydroxy group of 3-methyl catechol, OH3TOL, and OH4TOL is assumed to form an intermediate that then reacts with $NO_2$ to from a nitro compound. Under low-NO conditions, there is no loss process for this intermediate in the kinetic model or MCM v3.1.1. In experiments 1 and 2, after all injections were complete, lights on was

delayed for 2.5 h to estimate the wall loss of *o*-cresol. Wall loss of all other compounds is explained in section 4.2.1 in the main text.

[Figure]

**Figure S5.** Linear fit to natural log of wall deposition rate constant versus natural log of C* used to estimate wall deposition of compounds that cannot be directly measured.

**Table S3:** Reactions and reaction rate constants added to chemistry in MCM v3.3.1 to test proposed chemistry.

| New Reaction | New Reaction Rate [a] | Assumptions |
|---|---|---|
| **Version 1 – All reactions in MCM v 3.3.1 and those listed below.** | | |
| CRESOL → wall | 9.4 x $10^{-7}$ $s^{-1}$ | Measured in this study |
| MCATECHOL → wall | 2.5 x $10^{-6}$ $s^{-1}$ | Measured in this study |
| **Version 2 – All reactions in Version 1 and those listed below.** | | |
| TOL1OHNO2 + hv → products | 1.73 x $10^{-4}$ $s^{-1}$ | Assume similar to |
| MNCATECH + hv → products | 1.73 x $10^{-4}$ $s^{-1}$ | 6-methyl-2-nitrophenol (Bejan et al., 2007) |
| **Version 3 – All reactions in Version 2 and those listed below.** | | |
| CRESOL + OH → BCRESOL | 4.65 x $10^{-11}$ * 0.2 * 0.65 | Assume missing products from |
| | | Olariu et al. (2002) from bicyclic pathway. |
| MCATECHOL + OH → MCATEC1O | 2.0 x $10^{-10}$ * 0.07 | Assume same as *o*-cresol |
| MCATECHOL + OH → OHMBQN | 2.0 x $10^{-10}$ * 0.07 | oxidation from MCM v3.3.1 |
| MCATECHOL + OH → OH3TOL | 2.0 x $10^{-10}$ * 0.73 | and (Olariu et al., 2002) |
| MCATECHOL + OH → BMCATECHOL | 2.0 x $10^{-10}$ * 0.13 | |
| OH3TOL + OH → OH3TOL1O | 2.5 x $10^{-10}$ * 0.07 | Assume same as *o*-cresol oxidation |

**Table S3:** Reactions and reaction rate constants added to chemistry in MCM v3.3.1 to test proposed chemistry.

| New Reaction | New Reaction Rate [a] | Assumptions |
|---|---|---|
| OH3TOL + OH → OH2MBQN | $2.5 \times 10^{-10} * 0.07$ | from MCM v3.3.1 and (Olariu et al., 2002, 2000). |
| OH3TOL + OH → OH4TOL | $2.5 \times 10^{-10} * 0.73$ | Increased reaction rate constant due to additional |
| OH3TOL + OH → BOH3TOL | $2.5 \times 10^{-10} * 0.13$ | OH group to hard sphere collision rate limit. |
| OH4TOL + OH → OH4TOL1O | $2.5 \times 10^{-10} * 0.07$ | Assume same as o-cresol oxidation |
| OH4TOL + OH → OH3MBQN | $2.5 \times 10^{-10} * 0.07$ | from MCM v3.3.1 and Olariu et al. (2002, 2000). |
| OH4TOL + OH → OH5TOL | $2.5 \times 10^{-10} * 0.73$ | Increased reaction rate constant due to additional |
| OH4TOL + OH → BOH4TOL | $2.5 \times 10^{-10} * 0.13$ | OH group to hard sphere collision rate limit. |
| BCRESOL + OH → products | $5.44 \times 10^{-11}$ | Assume same as |
| BMCATECHOL + OH → products | $5.44 \times 10^{-11}$ | C5CO14OH from MCM v 3.3.1 |
| BOH3TOL + OH → products | $5.44 \times 10^{-11}$ | |
| BOH4TOL + OH → products | $5.44 \times 10^{-11}$ | |
| OHMBQN + OH → products | $2.3 \times 10^{-11}$ | Assume same as |
| OH2MBQN + OH → products | $2.3 \times 10^{-11}$ | PTLQONE from MCM v 3.3.1 |
| OH3TOL1O + NO$_2$ → products | $2.08 \times 10^{-12}$ | Assume same as |
| OH4TOL1O + NO$_2$ → products | $2.08 \times 10^{-12}$ | MCATEC1O from MCM v3.3.1 |
| HOC6H4NO2 + $hv$ → products | $6.13 \times 10^{-5}$ s$^{-1}$ | Based on 2-nitrophenol measured by Bardini (2006) reported by Chen et al. (2011) |
| C6H5O + NO → C6H5O(NO) | $1.65 \times 10^{-12}$ | Berho et al. (1998) |
| C6H5O(NO) + OH → products | $9.0 \times 10^{-13}$ | Assume same as |
| C6H5O(NO) + NO$_3$ → products | $9.0 \times 10^{-14}$ | HOC6H4NO2 from MCM v3.3.1 |
| OH3TOL → wall | $2.1 \times 10^{-5}$ s$^{-1}$ | Measured in this study |
| OH4TOL → wall | $7.9 \times 10^{-5}$ s$^{-1}$ | Estimated in this study |
| OH5TOL → wall | $5.0 \times 10^{-4}$ s$^{-1}$ | Estimated in this study |
| OHMBQN → wall | $9.6 \times 10^{-6}$ s$^{-1}$ | Measured in this study |
| OH2MBQN → wall | $2.0 \times 10^{-5}$ s$^{-1}$ | Measured in this study |
| OH3MBQN → wall | $1.2 \times 10^{-4}$ s$^{-1}$ | Estimated in this study |

[a] Reaction rate units are cm$^3$ molec$^{-1}$ s$^{-1}$ unless otherwise noted.

**Table S4.** Abbreviations used in Table S3

| Abbreviation | Description |
| --- | --- |
| BCRESOL | Tracer for products from the bicyclic intermediate pathway from cresol oxidation. |
| BMCATECHOL | Tracer for products from the bicyclic intermediate pathway from methyl catechol. |
| BOH3TOL | Tracer for products from the bicyclic intermediate pathway from trihydroxy toluene. |
| BOH4TOL | Tracer for products from the bicyclic intermediate pathway from tetrahydroxy toluene. |
| C5CO14OH | Acetyl acrylic acid (one of the bicyclic intermediate pathway products from o-cresol oxidation in MCM). |
| CRESOL | Cresol |
| HOC6H4NO2 | Nitrophenol |
| MCATEC1O | Product from H-abstraction of OH group of methyl catechol |
| MCATECHOL | Methyl catechol |
| MNCATECH | Nitro dihydroxy toluene |
| OH2MBQN | Dihydroxy methyl benzoquinone |
| OH3MBQN | Trihydroxy methyl benzoquinone |
| OH3TOL | Trihydroxy toluene |
| OH3TOL1O | Product from H-abstraction of OH group of trihydroxy toluene |
| OH3TOL1O | Product from H-abstraction of OH group of tetrahydroxy toluene |
| OH4TOL | Tetrahydroxy toluene |
| OH5TOL | Pentahydroxy toluene |
| OHMBQN | Hydroxy methyl benzoquinone |
| PTLQONE | Methyl benzoquinone (one of the bicyclic intermediate pathway products from cresol oxidation in MCM) |
| TOL1OHNO2 | Nitro hydroxy toluene |

**Table S5.** Chamber conditions based on kinetic model (Version 1)

| Expt # | VOC-OH adduct rxn (%) | | o-cresol rxn (%) | | 3-methyl catechol rxn (%) | | RO$_2$ Reaction Partner (%) | | |
|---|---|---|---|---|---|---|---|---|---|
| | O$_2$ | NO$_2$ | OH | NO$_3$ | OH | NO$_3$ | RO$_2$ | HO$_2$ | NO |
| 1 | 94 | 6 | >31 | <69 | >41 | <59 | ∼0 | <1 | >99 |
| 2 | 100 | 0 | 100 | 0 | 100 | 0 | <12 | >88 | ∼0 |
| 3 | 94 | 6 | >44 | <56 | >44 | <56 | ∼0 | <1 | >99 |
| 4 | 100 | 0 | 100 | 0 | 100 | 0 | <6 | >94 | ∼0 |
| 5 | >99.9 | <0.1 | >96 | <4 | >91 | <9 | ∼0 | <1 | >99 |
| 6 | 100 | 0 | 100 | 0 | 100 | 0 | <1 | >99 | ∼0 |
| 7 | >99.9 | <0.1 | NA | NA | ∼100 | ∼0 | ∼0 | <2 | >98 |
| 8 | 100 | 0 | NA | NA | 100 | 0 | No RO$_2$ forms in MCM | | |
| 9 | 100 | 0 | NA | NA | 100 | 0 | from low NO oxidation | | |
| 10 | 100 | 0 | NA | NA | 100 | 0 | of methyl catechol. | | |
| 11 | 100 | 0 | NA | NA | NA | NA | ∼16[a] | ∼84 [a] | ∼0 [a] |
| 12 | 97 | 3 | NA | NA | NA | NA | ∼0 | <4 | >96 |
| 13 | 100 | 0 | 100 | 0 | 100 | 0 | <18 | >82 | ∼0 |
| 14 | 90 | 10 | >20 | <80 | >34 | <66 | ∼0 | <1 | >99 |
| 15 | >99.9 | <0.1 | >98 | <2 | >94 | <6 | ∼0 | <1 | >99 |

[a] Throughout most of the experiment, the peroxy radical distribution was that stated. However, over the first hour there was exponential convergence to these steady state values from RO$_2$ + RO$_2$ = 100% and RO$_2$ + HO$_2$ = 0%.

**S3   DART-MS Analysis Details and Product Identification**

**S3.1   DART-MS Analysis Details**

A mass calibrant and an independent quality assurance/quality control (QA/QC) compound were run with each sample set to ensure mass accuracy to within 5 mDa. The mass calibrant used for positive mode was polyethylene glycol (average molecular weight of 600 amu, PEG-600; Acros Organics, Geel, Belgium), which was dissolved in methanol. The independent QA/QC compound used is reserpine, which was purchased from Sigma-Aldrich and diluted in methanol.

Tweezers were used to introduce the samples into the DART gas stream. Before analysis, the tweezers were rinsed with acetone, and were introduced into the gas stream to vaporize any contaminants. A strip (∼1 cm) was cut from each sample substrate for testing. The cutting was tested in triplicate, with each sampling being from a different are of the substrate.

In these studies, a solution of PEG-600 (50 $\mu$L in 10 mL of methanol) was used to calibrate (61-679 Da) the mass spectrometer for each run. Acceptable calibration was determined if the calibration Mass Center software produced a residual value of >9 x $10^{-12}$. To ensure proper calibration, a solution of reserpine (5 mg in 10 mL of methanol) was analyzed subsequent

to the PEG-600 in every sample run. Calibration was deemed sufficient if the m/z of reserpine fell within $\pm$ 0.005 Da of the theoretical value (609.281 Da).

The instrument used was a JEOL (Tokyo, Japan) AccuTOF™ mass spectrometer (JMS-T100LC) coupled with an IonSense (Saugus, MA, USA) DART® source. Ultra-pure helium was used as the ionizing gas with a flow rate of 1.75 L min$^{-1}$. For all analyses, the DART® source was set to a needle voltage of $\pm$3.5 kV. Electrode 1 and electrode 2 voltages were both set to $\pm$150 V. Mass spectrometer settings include: an orifice 1 voltage of $\pm$20 V, orifice 2 voltage of $\pm$5 V, a ring lens voltage of $\pm$5 V, a peaks voltage of 1500 V, a mass range of 50 – 1500 m/z at 0.5 seconds per scan. A helium gas stream temperature of 325 °C was also employed.

**S3.2  DART-MS Product Identification**

Best available knowledge was used to assign the compounds displayed in Tables S6, S7, and S8. The smaller compounds could be fragmentation products. $C_xH_yNO$ and $C_xH_yNO_2$ were assumed to be amines. These products could also be small nitro or nitroso compounds or fragmentation products of nitrates. Products that appeared to be fragmentation products (i.e., reasonable structures could not be drawn) were excluded from the list. The structure of each compound was necessary to estimate the vapor pressure. The most probable dominant isomer was selected in all cases, but there are likely many additional structural isomers that form as well. The abundances reported in Tables S6, S7, and S8 are not meant to be used quantitatively due to uncertainties in the vapor pressure estimation methods and centroid fitting algorithm. Often each m/z contained many over-lapping peaks and corrections were not made for isoptope effects.

**Table S6.** DART-MS data from low-NO toluene oxidation (experiment 13).

| m/z (+) (Da) | Intensity (A.U.) | C | H | N | O | $\Delta$ (mDa) [a] | Smiles | Est. VP (atm) [b] | Abundance ($R_f$) |
|---|---|---|---|---|---|---|---|---|---|
| 69.067377 | 6671.45 | 5 | 8 | 0 | 0 | 3.05 | C=CC=CC | 6.49E-01 (E) | 3.42E-12 |
| 71.046739 | 11123.71 | 4 | 6 | 0 | 1 | 2.95 | CC=CC=O | 8.30E-02 (E) | 4.46E-11 |
| 71.081281 | 3477.48 | 5 | 10 | 0 | 0 | 4.79 | C=CCCC | 6.49E-01 (E) | 1.78E-12 |
| 73.064072 | 4280.44 | 4 | 8 | 0 | 1 | 1.27 | CCCC=O | 1.27E-01 (E) | 1.12E-11 |
| 75.04371 | 5382.00 | 3 | 6 | 0 | 2 | 0.89 | CC(CO)=O | 6.92E-03 (E) | 2.59E-10 |
| 80.048339 | 155429.63 | 5 | 5 | 1 | 0 | 1.69 | C1=CC=CC=N1 | 3.46E-02 (N) | 1.50E-09 |
| 83.082762 | 4393.36 | 6 | 10 | 0 | 0 | 3.31 | C=CCCC=C | 2.12E-01 (E) | 6.89E-12 |
| 85.025484 | 4539.76 | 4 | 4 | 0 | 2 | 3.47 | O=CC=CC=O | 3.42E-03 (E) | 4.41E-10 |
| 87.039802 | 7748.51 | 4 | 6 | 0 | 2 | 4.80 | O=CCCC=O | 6.19E-03 (E) | 4.16E-10 |
| 97.026419 | 6180.71 | 5 | 4 | 0 | 2 | 2.53 | O=CC1=CC=CO1 | 2.96E-03 (N) | 6.95E-10 |
| 97.055251 | 2871.86 | 6 | 8 | 0 | 1 | 10.09 | CC(C=CC=C)=O | 8.87E-03 (E) | 1.08E-10 |
| 97.101391 | 2954.88 | 7 | 12 | 0 | 0 | 0.33 | CC1C=CCCC1 | 6.93E-02 (E) | 1.42E-11 |
| 99.043366 | 9545.71 | 5 | 6 | 0 | 2 | 1.24 | O=C(C)C=CC=O | 1.12E-03 (E) | 2.84E-09 |
| 99.072496 | 3758.17 | 6 | 10 | 0 | 1 | 8.49 | CC(C=CCC)=O | 8.87E-03 (E) | 1.41E-10 |
| 101.057523 | 4501.87 | 5 | 8 | 0 | 2 | 2.73 | O=C(C)CCC=O | 2.02E-03 (E) | 7.40E-10 |
| 109.035545 | 3507.03 | 6 | 4 | 0 | 2 | -6.59 | O=C1C=CC(C=C1)=O | 2.48E-05 (E) | 4.69E-08 |
| 109.096678 | 4467.03 | 8 | 12 | 0 | 0 | 5.05 | C=CC=CC=CCC | 2.27E-02 (E) | 6.55E-11 |
| 111.043476 | 6721.86 | 6 | 6 | 0 | 2 | 1.13 | O=CC1=CC=C(O1)C | 1.25E-03 (N) | 1.79E-09 |
| 111.11751 | 3129.72 | 8 | 14 | 0 | 0 | -0.13 | CCC=CC=CCC | 2.27E-02 (E) | 4.59E-11 |
| 113.019938 | 3471.30 | 5 | 4 | 0 | 3 | 3.93 | O=C1C(C)=CC(O1)=O | 7.03E-04 (E) | 1.64E-09 |
| 113.05728 | 4021.97 | 6 | 8 | 0 | 2 | 2.97 | O=CCCC=CC=O | 4.92E-04 (E) | 2.72E-09 |
| 115.038947 | 6483.21 | 5 | 6 | 0 | 3 | 0.57 | O=C(C)C=CC(O)=O | 6.57E-06 (E) | 3.28E-07 |
| 115.064062 | 2955.96 | 6 | 10 | 0 | 2 | 11.84 | O=CCCCCC=O | 6.62E-04 (E) | 1.49E-09 |
| 127.039667 | 5945.22 | 6 | 6 | 0 | 3 | -0.15 | O=C(C)C=CC(C=O)=O | 1.64E-04 (E) | 1.21E-08 |
| 127.112254 | 4155.67 | 8 | 14 | 0 | 1 | 0.04 | O=CCCCCC=CC | 1.45E-03 (E) | 9.55E-10 |
| 129.053327 | 4254.59 | 6 | 8 | 0 | 3 | 1.84 | O=C(C=CC(O)C=O)C | 6.36E-05 (E) | 2.22E-08 |
| 139.034538 | 5275.37 | 7 | 6 | 0 | 3 | 4.98 | CC1=CC(C=C(O)C1=O)=O | 5.26E-07 (E) | 3.34E-06 |
| 141.050361 | 3122.55 | 7 | 8 | 0 | 3 | 4.81 | CC1=CC=C(O)C(O)=C1O | 5.97E-08 (N) | 1.74E-05 |
| 155.034837 | 3399.08 | 7 | 6 | 0 | 4 | -0.40 | CC1=CC(C(O)=C(O)C1=O)=O | 8.26E-09 (E) | 1.37E-04 |
| 157.045695 | 8025.67 | 7 | 8 | 0 | 4 | 4.39 | CC1=CC(O)=C(O)C(O)=C1O | 3.28E-10 (N) | 8.13E-03 |
| 161.091424 | 2838.46 | 11 | 12 | 0 | 1 | 5.22 | O=CC=CC=CC=CC=CC=C | 3.32E-05 (E) | 2.85E-08 |
| 173.044149 | 3338.73 | 7 | 8 | 0 | 5 | 0.85 | CC1=C(O)C(O)=C(C(O)=C1O)O | 1.12E-12 (N) | 9.92E-01 |
| 203.10019 | 3898.22 | 13 | 14 | 0 | 2 | 7.01 | O=CC=CC=CC=CC=CC=CC(C)=O | 1.46E-07 (E) | 8.87E-06 |

[a] The difference between the measured and proposed compound exact mass. [b] Est. VP = Estimated vapor pressure. Estimation Method in parenthesis: E = EVAPORATION method and N = Nannoolal method.

**Table S7:** DART-MS data from high-NO *o*-cresol oxidation (experiment 15).

| m/z (+) (Da) | Intensity [a] | C | H | N | O | Δ [b] (mDa) | Smiles | Est. VP (atm) [c] | Abundance (R$_f$) |
|---|---|---|---|---|---|---|---|---|---|
| 69.06738 | 4799.51 | 5 | 8 | 0 | 0 | 3.05 | C=CC=CC | 6.49E-01 (E) | 1.17E-11 |
| 71.04674 | 7360.97 | 4 | 6 | 0 | 1 | 2.95 | CC=CC=O | 8.30E-02 (E) | 1.40E-10 |
| 73.06407 | 3055.72 | 4 | 8 | 0 | 1 | 1.27 | CCCC=O | 1.27E-01 (E) | 3.82E-11 |
| 75.04371 | 3495.06 | 3 | 6 | 0 | 2 | 0.89 | CC(CO)=O | 6.92E-03 (E) | 7.99E-10 |
| 76.0358 | 2830.61 | 2 | 5 | 1 | 2 | 4.06 | OCC(N)=O | 5.43E-08 (N) | 8.25E-05 |
| 80.04834 | 2555.95 | 5 | 5 | 1 | 0 | 1.69 | C1=CC=CC=N1 | 3.46E-02 (N) | 1.17E-10 |
| 81.0676 | 2116.91 | 6 | 8 | 0 | 0 | 2.83 | C=CC=CC=C | 2.12E-01 (E) | 1.58E-11 |
| 83.08276 | 2977.45 | 6 | 10 | 0 | 0 | 3.31 | C=CCCC=C | 2.12E-01 (E) | 2.22E-11 |
| 85.02548 | 2862.51 | 4 | 4 | 0 | 2 | 3.47 | O=CC=CC=O | 3.42E-03 (E) | 1.32E-09 |
| 85.06327 | 2015.80 | 5 | 8 | 0 | 1 | 2.07 | CC(C=CC)=O | 2.71E-02 (E) | 1.17E-10 |
| 87.04526 | 4217.44 | 4 | 6 | 0 | 2 | -0.66 | O=CCCC=O | 6.19E-03 (E) | 1.08E-09 |
| 94.06261 | 2373.32 | 6 | 7 | 1 | 0 | 3.06 | N1C=CC=CC=C1 | 2.04E-02 (N) | 1.84E-10 |
| 95.08159 | 1934.09 | 7 | 10 | 0 | 0 | 4.49 | CC1C=CCC=C1 | 6.93E-02 (E) | 4.41E-11 |
| 97.02642 | 4443.40 | 5 | 4 | 0 | 2 | 2.53 | O=CC1=CC=CO1 | 2.96E-03 (N) | 2.38E-09 |
| 97.06102 | 3408.60 | 6 | 8 | 0 | 1 | 4.32 | CC(C=CC=C)=O | 8.87E-03 (E) | 6.08E-10 |
| 97.09562 | 2367.86 | 7 | 12 | 0 | 0 | 6.10 | CC1C=CCCC1 | 6.93E-02 (E) | 5.40E-11 |
| 98.06128 | 2115.28 | 5 | 7 | 1 | 1 | -0.69 | NC(C=CC=C)=O | 1.44E-06 (N) | 2.33E-06 |
| 99.04337 | 5939.65 | 5 | 6 | 0 | 2 | 1.24 | O=C(C)C=CC=O | 1.12E-03 (E) | 8.40E-09 |
| 99.07832 | 3689.73 | 6 | 10 | 0 | 1 | 2.67 | CC(C=CCC)=O | 8.87E-03 (E) | 6.58E-10 |
| 100.0362 | 2821.45 | 4 | 5 | 1 | 2 | 3.65 | O=CC=CC(N)=O | 2.02E-07 (N) | 2.21E-05 |
| 101.0222 | 2194.10 | 4 | 4 | 0 | 3 | 1.65 | O=CC=CC(O)=O | 2.01E-05 (E) | 1.73E-07 |
| 101.0575 | 3823.68 | 5 | 8 | 0 | 2 | 2.73 | O=C(C)CCC=O | 2.02E-03 (E) | 2.99E-09 |
| 102.0545 | 2971.25 | 4 | 7 | 1 | 2 | 1.54 | NCC=CC(O)=O | 2.92E-05 (N) | 1.61E-07 |
| 104.0332 | 5611.42 | 3 | 5 | 1 | 3 | 1.60 | CC=CON(=O)=O | 2.60E-02 (E) | 3.42E-10 |
| 109.1028 | 3360.34 | 8 | 12 | 0 | 0 | -1.07 | C=CC=CC=CCC | 2.27E-02 (E) | 2.34E-10 |
| 111.0435 | 4461.87 | 6 | 6 | 0 | 2 | 1.13 | O=CC1=CC=C(O1)C | 1.25E-03 (N) | 5.64E-09 |
| 111.1175 | 2507.94 | 8 | 14 | 0 | 0 | -0.13 | CCC=CC=CCC | 2.27E-02 (E) | 1.75E-10 |
| 113.0262 | 1862.29 | 5 | 4 | 0 | 3 | -2.29 | O=C1C(C)=CC(O1)=O | 7.03E-04 (E) | 4.19E-09 |
| 113.0573 | 3145.91 | 6 | 8 | 0 | 2 | 2.97 | O=CCCC=CC=O | 4.92E-04 (E) | 1.01E-08 |
| 114.0553 | 2382.57 | 5 | 7 | 1 | 2 | 0.17 | C=C(C=CC(O)=O)N | 1.45E-05 (N) | 2.59E-07 |
| 115.0389 | 3284.56 | 5 | 6 | 0 | 3 | 0.57 | O=C(C)C=CC(O)=O | 6.57E-06 (E) | 7.90E-07 |
| 115.0703 | 2005.33 | 6 | 10 | 0 | 2 | 5.56 | O=CCCCCC=O | 6.62E-04 (E) | 4.79E-09 |
| 118.0469 | 2527.17 | 4 | 7 | 1 | 3 | 3.51 | CCC=CON(=O)=O | 8.50E-03 (E) | 4.71E-10 |
| 120.0524 | 3185.49 | 4 | 9 | 1 | 3 | 13.66 | CCCCON(=O)=O | 8.50E-03 (E) | 5.93E-10 |
| 126.0519 | 1896.52 | 6 | 7 | 1 | 2 | 3.58 | OC1=CN=C(C)C(O)=C1 | 1.36E-05 (N) | 2.20E-07 |

**Table S7:** DART-MS data from high-NO *o*-cresol oxidation (experiment 15).

| m/z (+) (Da) | Intensity [a] | C | H | N | O | Δ [b] (mDa) | Smiles | Est. VP (atm) [c] | Abundance ($R_f$) |
|---|---|---|---|---|---|---|---|---|---|
| 127.0397 | 3814.74 | 6 | 6 | 0 | 3 | -0.15 | O=C(C)C=CC(C=O)=O | 1.64E-04 (E) | 3.68E-08 |
| 127.0661 | 2199.51 | 7 | 10 | 0 | 2 | 9.84 | O=C(C)CC=CCC=O | 2.16E-04 (E) | 1.61E-08 |
| 127.1123 | 3306.85 | 8 | 14 | 0 | 1 | 0.04 | O=CCCCCC=CC | 1.45E-03 (E) | 3.61E-09 |
| 128.071 | 1864.39 | 6 | 9 | 1 | 2 | 0.14 | O=C(C=CC=CCN)O | 3.46E-06 (N) | 8.52E-07 |
| 129.0533 | 2463.10 | 6 | 8 | 0 | 3 | 1.84 | O=C(C=CC(O)C=O)C | 6.36E-05 (E) | 6.12E-08 |
| 130.0527 | 1871.23 | 5 | 7 | 1 | 3 | -2.32 | O=N(OC=CC=CC)=O | 2.78E-03 (E) | 1.07E-09 |
| 139.0414 | 2114.31 | 7 | 6 | 0 | 3 | -1.92 | CC1=CC(C=C(O)C1=O)=O | 5.26E-07 (E) | 6.36E-06 |
| 142.0463 | 2154.17 | 6 | 7 | 1 | 3 | 4.15 | C=CC=CC(ON(=O)=O)=C | 9.08E-04 (E) | 3.75E-09 |
| 154.0524 | 1956.21 | 7 | 7 | 1 | 3 | -2.01 | OC1=C(N(=O)=O)C=CC=C1C | 1.77E-05 (N) | 1.75E-07 |
| 155.0348 | 3607.35 | 7 | 6 | 0 | 4 | -0.40 | CC1=CC(C(O)=C(O)C1=O)=O | 8.26E-09 (E) | 6.91E-04 |
| 157.0457 | 1941.16 | 7 | 8 | 0 | 4 | 4.39 | CC1=CC(O)=C(O)C(O)=C1O | 3.28E-10 (N) | 9.36E-03 |
| 267.1658 | 2870.33 | 15 | 22 | 0 | 4 | -6.13 | OC(OC1=C(O)C(O)=CC=C1C)CCCCC=CC [c] | 4.59E-12 (N) | 9.90E-01 |

[a] (A.U.) [b] The difference between the measured and proposed compound exact mass. [c] Est. VP = Estimated vapor pressure. Estimation Method in parenthesis: E = EVAPORATION method, and N = Nannoolal method. [c] Smiles in table is that of the structure predicted to form. Vapor pressure method could not estimate the vapor pressure of this structure so a very similar structure was used instead (OC(C(O)=CC=C1C)=C1OCC(O)CCCC=CC).

**Table S8:** DART-MS data from high-NO toluene oxidation (experiment 14).

| m/z (+) (Da) | Intensity [a] | C | H | N | O | Δ [b] (mDa) | Smiles | Est. VP (atm) [c] | Abundance ($R_f$) |
|---|---|---|---|---|---|---|---|---|---|
| 61.026497 | 5540.57 | 2 | 4 | 0 | 2 | 2.46 | CC(O)=O | 4.49E-03 (E) | 7.02E-10 |
| 69.067377 | 1410.70 | 5 | 8 | 0 | 0 | 3.05 | C=CC=CC | 6.49E-01 (E) | 1.24E-12 |
| 76.035796 | 3623.11 | 2 | 5 | 1 | 2 | 4.06 | OCC(N)=O | 5.43E-08 (N) | 3.80E-05 |
| 80.048339 | 1641.73 | 5 | 5 | 1 | 0 | 1.69 | C1=CC=CC=N1 | 3.46E-02 (N) | 2.71E-11 |
| 83.082762 | 1684.25 | 6 | 10 | 0 | 0 | 3.31 | C=CCCC=C | 2.12E-01 (E) | 4.52E-12 |
| 85.025484 | 2836.08 | 4 | 4 | 0 | 2 | 3.47 | O=CC=CC=O | 3.42E-03 (E) | 4.72E-10 |
| 87.007038 | 2179.84 | 3 | 2 | 0 | 3 | 1.18 | O=CC(C=O)=O | 7.03E-02 (E) | 1.77E-11 |
| 87.039802 | 6440.08 | 4 | 6 | 0 | 2 | 4.80 | O=CCCC=O | 6.19E-03 (E) | 5.93E-10 |
| 90.013837 | 2080.44 | 2 | 3 | 1 | 3 | 5.28 | C=CON(=O)=O | 7.95E-02 (E) | 1.49E-11 |
| 90.047162 | 1148.22 | 3 | 7 | 1 | 2 | 8.34 | OC(C)C(N)=O | 2.13E-07 (N) | 3.07E-06 |
| 94.06261 | 7800.17 | 6 | 7 | 1 | 0 | 3.06 | N1C=CC=CC1 | 2.04E-02 (N) | 2.18E-10 |

**Table S8:** DART-MS data from high-NO toluene oxidation (experiment 14).

| m/z (+) (Da) | Intensity [a] | C | H | N | O | $\Delta$ [b] (mDa) | Smiles | Est. VP (atm) [c] | Abundance ($R_f$) |
|---|---|---|---|---|---|---|---|---|---|
| 95.053048 | 1261.49 | 6 | 6 | 0 | 1 | -3.36 | OC1=CC=CC=C1 | 1.44E-03 (N) | 4.98E-10 |
| 95.081585 | 1364.08 | 7 | 10 | 0 | 0 | 4.49 | CC1C=CCC=C1 | 6.93E-02 (E) | 1.12E-11 |
| 97.026419 | 6227.32 | 5 | 4 | 0 | 2 | 2.53 | O=CC1=CC=CO1 | 2.96E-03 (N) | 1.20E-09 |
| 97.061018 | 4637.55 | 6 | 8 | 0 | 1 | 4.32 | CC(C=CC=C)=O | 8.87E-03 (E) | 2.98E-10 |
| 98.06128 | 1349.16 | 5 | 7 | 1 | 1 | -0.69 | NC(C=CC=C)=O | 1.44E-06 (N) | 5.35E-07 |
| 99.043366 | 11466.81 | 5 | 6 | 0 | 2 | 1.24 | O=C(C)C=CC=O | 1.12E-03 (E) | 5.84E-09 |
| 99.089977 | 2449.13 | 6 | 10 | 0 | 1 | -8.99 | CC(C=CCC)=O | 8.87E-03 (E) | 1.57E-10 |
| 100.042055 | 2453.04 | 4 | 5 | 1 | 2 | -2.20 | O=CC=CC(N)=O | 2.02E-07 (N) | 6.91E-06 |
| 100.071332 | 1379.97 | 5 | 9 | 1 | 1 | 4.91 | NC(C=CCC)=O | 2.52E-06 (N) | 3.12E-07 |
| 101.022218 | 2932.23 | 4 | 4 | 0 | 3 | 1.65 | O=CC=CC(O)=O | 2.01E-05 (E) | 8.30E-08 |
| 101.051638 | 1499.95 | 5 | 8 | 0 | 2 | 8.62 | O=C(C)CCC=O | 2.02E-03 (E) | 4.22E-10 |
| 102.054463 | 1439.00 | 4 | 7 | 1 | 2 | 1.54 | NCC=CC(O)=O | 2.92E-05 (N) | 2.81E-08 |
| 102.089947 | 2487.60 | 5 | 11 | 1 | 1 | 1.94 | NC(CCCC)=O | 3.50E-06 (N) | 4.05E-07 |
| 103.03847 | 1574.86 | 4 | 6 | 0 | 3 | 1.05 | CC(C(C=O)O)=O | 2.00E-03 (E) | 4.48E-10 |
| 104.033168 | 2667.65 | 3 | 5 | 1 | 3 | 1.60 | CC=CON(=O)=O | 2.60E-02 (E) | 5.85E-11 |
| 105.014648 | 2141.85 | 3 | 4 | 0 | 4 | 4.14 | O=C(O)C(CO)=O | 2.22E-06 (E) | 5.49E-07 |
| 109.096678 | 3357.59 | 8 | 12 | 0 | 0 | 5.05 | C=CC=CC=CCC | 2.27E-02 (E) | 8.44E-11 |
| 110.058713 | 1424.19 | 6 | 7 | 1 | 1 | 1.88 | OC1=CC=CN=C1C | 1.53E-03 (N) | 5.29E-10 |
| 111.043476 | 4684.00 | 6 | 6 | 0 | 2 | 1.13 | O=CC1=CC=C(O1)C | 1.25E-03 (N) | 2.13E-09 |
| 112.038821 | 2037.51 | 5 | 5 | 1 | 2 | 1.03 | OC1=CC(O)=CN=C1 | 5.32E-05 (N) | 2.18E-08 |
| 113.026161 | 2657.15 | 5 | 4 | 0 | 3 | -2.29 | O=C1C(C)=CC(O1)=O | 7.03E-04 (E) | 2.15E-09 |
| 113.05728 | 3740.34 | 6 | 8 | 0 | 2 | 2.97 | O=CCCC=CC=O | 4.92E-04 (E) | 4.33E-09 |
| 114.055338 | 2438.47 | 5 | 7 | 1 | 2 | 0.17 | C=C(C=CC(O)=O)N | 1.45E-05 (N) | 9.56E-08 |
| 115.038947 | 7677.84 | 5 | 6 | 0 | 3 | 0.57 | O=C(C)C=CC(O)=O | 6.57E-06 (E) | 6.65E-07 |
| 116.033084 | 1330.37 | 4 | 5 | 1 | 3 | 1.68 | C=CC=CON(=O)=O | 8.50E-03 (E) | 8.92E-11 |
| 116.064614 | 1348.08 | 5 | 9 | 1 | 2 | 6.54 | CC(N)C=CC(O)=O | 1.87E-05 (N) | 4.12E-08 |
| 117.050497 | 2483.76 | 5 | 8 | 0 | 3 | 4.67 | O=C(C)CCC(O)=O | 1.00E-05 (E) | 1.41E-07 |
| 118.046909 | 1887.46 | 4 | 7 | 1 | 3 | 3.51 | CCC=CON(=O)=O | 8.50E-03 (E) | 1.27E-10 |
| 123.046863 | 1686.79 | 7 | 6 | 0 | 2 | -2.26 | CC1=CC(C=CC1=O)=O | 8.12E-06 (E) | 1.18E-07 |
| 125.06148 | 1442.25 | 7 | 8 | 0 | 2 | -1.23 | CC1=CC=CC(O)=C1O | 6.77E-06 (N) | 1.21E-07 |
| 126.051919 | 2178.90 | 6 | 7 | 1 | 2 | 3.58 | OC1=CN=C(C)C(O)=C1 | 1.36E-05 (N) | 9.10E-08 |
| 127.039667 | 4187.31 | 6 | 6 | 0 | 3 | -0.15 | O=C(C)C=CC(C=O)=O | 1.64E-04 (E) | 1.46E-08 |
| 127.112254 | 3174.50 | 8 | 14 | 0 | 1 | 0.04 | O=CCCCCC=CC | 1.45E-03 (E) | 1.25E-09 |
| 128.03127 | 2068.51 | 5 | 5 | 1 | 3 | 3.50 | O=N(C1=CC=C(C)O1)=O | 1.93E-03 (N) | 6.09E-10 |
| 129.053327 | 3905.59 | 6 | 8 | 0 | 3 | 1.84 | O=C(C=CC(O)C=O)C | 6.36E-05 (E) | 3.50E-08 |

**Table S8:** DART-MS data from high-NO toluene oxidation (experiment 14).

| m/z (+) (Da) | Intensity [a] | C | H | N | O | $\Delta$ [b] (mDa) | Smiles | Est. VP (atm) [c] | Abundance (R$_f$) |
|---|---|---|---|---|---|---|---|---|---|
| 130.052743 | 1472.64 | 5 | 7 | 1 | 3 | -2.32 | O=N(OC=CC=CC)=O | 2.78E-03 (E) | 3.02E-10 |
| 131.035911 | 1919.85 | 5 | 6 | 0 | 4 | -1.48 | OC(C(O)=C(C=O)C)=O | 1.09E-07 (E) | 1.00E-05 |
| 131.062715 | 1156.33 | 6 | 10 | 0 | 3 | 8.10 | O=C(C)CCC(C=O)O | 4.99E-05 (E) | 1.32E-08 |
| 133.047111 | 1311.02 | 5 | 8 | 0 | 4 | 2.97 | OC(C(C(C)C=O)O)=O | 1.22E-01 (E) | 6.13E-12 |
| 138.049261 | 2335.43 | 7 | 7 | 1 | 2 | 6.24 | CC1=C(N(=O)=O)C=CC=C1 | 2.18E-04 (N) | 6.12E-09 |
| 139.034538 | 2994.57 | 7 | 6 | 0 | 3 | 4.98 | CC1=CC(C=C(O)C1=O)=O | 5.26E-07 (E) | 3.24E-06 |
| 140.030245 | 2170.62 | 6 | 5 | 1 | 3 | 4.52 | OC1=CC=CC=C1N(=O)=O | 8.71E-05 (N) | 1.42E-08 |
| 141.050361 | 5078.50 | 7 | 8 | 0 | 3 | 4.81 | CC1=CC=C(O)C(O)=C1O | 5.97E-08 (N) | 4.85E-05 |
| 142.046271 | 3022.13 | 6 | 7 | 1 | 3 | 4.15 | C=CC=CC(ON(=O)=O)=C | 9.08E-04 (E) | 1.90E-09 |
| 143.031682 | 2933.82 | 6 | 6 | 0 | 4 | 2.75 | O=C(C)C=CC(C(O)=O)=O | 5.77E-07 (E) | 2.90E-06 |
| 145.047972 | 2291.30 | 6 | 8 | 0 | 4 | 2.11 | O=C(C=CC(O)C(O)=O)C | 3.56E-08 (E) | 3.66E-05 |
| 152.068107 | 1146.80 | 8 | 9 | 1 | 2 | 3.05 | NC(C=CC=CC=CC=O)=O | 4.05E-09 (N) | 1.61E-04 |
| 154.045164 | 2966.87 | 7 | 7 | 1 | 3 | 5.25 | OC1=C(N(=O)=O)C=CC=C1C | 1.77E-05 (N) | 9.55E-08 |
| 155.034837 | 3800.38 | 7 | 6 | 0 | 4 | -0.40 | CC1=CC(C(O)=C(O)C1=O)=O | 8.26E-09 (E) | 2.62E-04 |
| 156.06424 | 1417.63 | 7 | 9 | 1 | 3 | 1.83 | CC=CC=CC=CON(=O)=O | 2.97E-04 (E) | 2.72E-09 |
| 157.045695 | 5036.75 | 7 | 8 | 0 | 4 | 4.39 | CC1=CC(O)=C(O)C(O)=C1O | 3.28E-10 (N) | 8.74E-03 |
| 158.044945 | 1517.74 | 6 | 7 | 1 | 4 | 0.39 | O=N(OC=CC(CC=C)=O)=O | 3.80E-05 (E) | 2.27E-08 |
| 159.062128 | 1321.64 | 7 | 10 | 0 | 4 | 3.61 | CC(C=CC(C(O)C=O)O)=O | 3.41E-07 (E) | 2.21E-06 |
| 161.046849 | 1846.47 | 6 | 8 | 0 | 5 | -1.85 | O=C(CO)C=CC(C(O)=O)O | 1.79E-10 (E) | 5.89E-03 |
| 170.046401 | 1573.59 | 7 | 7 | 1 | 4 | -1.07 | OC1=C(O)C(N(=O)=O)=CC=C1C | 2.01E-07 (N) | 4.46E-06 |
| 173.044149 | 1855.62 | 7 | 8 | 0 | 5 | 0.85 | CC1=C(O)C(O)=C(C(O)=C1O)O | 1.12E-12 (N) | 9.44E-01 |
| 174.069813 | 1324.07 | 7 | 11 | 1 | 4 | 6.82 | O=N(OC=CCCCC=O)=O | 1.90E-05 (E) | 3.98E-08 |
| 175.059781 | 1319.16 | 7 | 10 | 0 | 5 | 0.87 | O=C(C)C=CC(O)C(O)C(O)=O | 1.33E-10 (E) | 5.64E-03 |
| 177.157199 | 1399.51 | 9 | 20 | 0 | 3 | -8.13 | CCCCC(O)C(O)CCCO | 2.29E-09 (E) | 3.49E-04 |
| 178.069959 | 2175.62 | 6 | 11 | 1 | 5 | 1.59 | O=N(OCCCCC(O)=O)=O | 2.20E-07 (E) | 5.63E-06 |
| 223.064145 | 2037.80 | 11 | 10 | 0 | 5 | -3.50 | O=C(O)C=CC=CC=CC=CC(C(O)=O)=O | 3.35E-11 (E) | 3.46E-02 |

[a] (A.U.) [b] The difference between the measured and proposed compound exact mass. [b] Est. VP = Estimated vapor pressure. Estimation Method in parenthesis: E = EVAPORATION method, and N = Nannoolal method.

Other studies have reported structural isomers of the compounds listed in Table S6, S7, and S8 in the gas-phase and particle-phase from toluene SOA (Jang and Kamens, 2001; Sato et al., 2007). Peaks for $C_7H_8O_4$ and $C_7H_8O_5$ had the largest intensity in

**Figure S6.** Epoxide pathway oxidation mechanism under both low- and high-NO conditions as recommended by MCM v3.3.1

the particle-phase measurements in the study by Sato et al. (2007), but it should be noted that only 1% of the SOA constituents were quantified in that study. Both of these prior studies (Jang and Kamens, 2001; Sato et al., 2007) suggest that the compounds are ring-opening products not produced from the cresol pathway. Given the new evidence from the $CF_3O^-$ CIMS in this study, it is clear that these compounds are produced from the cresol pathway.

5      Products detected in the particle-phase by the DART-MS under $o$-cresol high NO conditions are shown in Figure S7. An oligomer product, $C_{15}H_{22}O_4$, is detected as one of the dominant products in $o$-cresol oxidation under high-NO conditions (Figure S7). It is possible this product forms from oligomerization of trihydroxy toluene and $C_8H_{14}O$ to form a hemiacetal.

[Figure]

**Figure S7.** Products detected by DART-MS in the particle phase during oxidation of *o*-cresol under high NO conditions (experiment 15) with boxes identifying the following types of compounds: polyols (black), methyl benzoquinone type compounds (magenta), decomposition products from the bicyclic intermediate pathway (cyan), products with more than 7 carbons (gold), and nitro compounds (green).

---

## Author Comment (AC2) · 13 Jan 2017

Response Review 2

Comment: "General This is a very interesting and impressive paper on the oxidation of cresol studied at the Caltech chamber. Generally, it contains a wealth of new findings extending the knowledge on aromatics oxidation considerably. Apparently, despite the study of aromatics oxidation over some decades now, this appears another field where the now available mass spectometric techniques allow for the identification of reaction products which have not been identified before, especially for low NO conditions - but see and consider the note below. On the other hand, the paper confirms a lot of findings as they are implemented tinto the MCM 3.3.1, many of them for high NO conditions which are expected to be met in regions where aromatic VOC emissions coincide with

elevated NO and NOx levels. It seems that especially for low NO conditions many new observations are made but the authors should include into their paper a discussion where conditions with low NO enabling prefence of ROO + HO2 over ROO + NO can really be met. The paper has the potential to deliver valuable changes and additions to chemical schemes such as the MCM 3.3.1. as it conains infomation of the formation of products in oxidation generations beyond one and two."

Response: Thanks for your helpful comments and suggestions. To address your main question (where/when is the chemistry described in this work most relevant?) and to put the work in broader context we have added an atmospheric relevance section (new Section 5) after the discussion section. Also we have added text to Section 3.1 and updated Figures 3 and 4 to better articulate that the main products from 3-methyl catechol oxidation are produced under both low- and high-NO conditions. We have attached as a supplement to the reviewer 1 response a pdf file including a comparison between the ACPD version and our edited version for ease of viewing changes.

———

Comment: "p3, line 27: While it is understandable to avoid further complications during the present study, it would be extremely interesting to do a similar study at higher RH and then see the coupling to aqueous chemistry"

Response: We agree and hope that future studies will expand this work to better understand the coupling to aqueous chemistry.

———

Comment: "p18, section 4.2.4: Maybe a reference can be given for the classic formation of the endo-peroxide ? I wonder if a intramolecular H-shift could occur in systems like the peroxyl / endoperoxides at the different stages of the mechanisms discussed so that a hydroperoxide might form from this and not only by + HO2 / - O2 ? Maybe it would be timeöy to discuss such possibility."

Response: Yes, a reference has been added for the classic formation of the endo-peroxide. Thank you for this suggestion. We have added reference to a possible H-shift that might occur prior to formation of the bicyclic intermediate peroxy radical in section 4.2.4 and section 4.1. Because this H-shift is suggested to be negligible for the phenol system from theoretical work by Xie et al., 2013, we also assume that products from this pathway will be negligible in this work. Given the large number of compounds detected in this work separating out possible products from the H-shift pathway from other pathways already discussed is speculative. Unfortunately, we cannot experimen-tally verify that this H-shift is minimal.

——

Comment: "Figure 3: The main additions here are on the catechol oxidation."

Response: Yes, this is true. The detected first-generation products from cresol oxida-tion are consistent with other studies. Identification of the second- and later- generation products from cresol oxidation is the focus of this work.

——

Comment: "Figure 4: I do not understand "See Figure 2 " written at the catechol up right in the scheme. Probabyl this should be 'Figure 3 ' ? Please explain / correct."

Response: We apologize for the confusion. This is a typo and has been fixed. Thank you for bringing this to our attention.

——

Comment: "References: I wonder if the paper PAN, Shan-Shan, and Li-Ming WANG. "The Atmospheric Oxidation Mechanism of o-Xylene Initiated by Hydroxyl Radicals." Acta Physico-Chimica Sinica 31.12 (2015): 2259-2268. should be mentioned because it contains remarkable recent considerations on xylene oxidation"

Response: We have referenced this paper in section 4.2.4 while responding to comment 3.